# JOINTIST: SIMULTANEOUS IMPROVEMENT OF MULTI-INSTRUMENT TRANSCRIPTION AND MUSIC SOURCE SEPARATION VIA JOINT TRAINING

## ABSTRACT

In this paper, we introduce Jointist, an instrument-aware multi-instrument framework that is capable of transcribing, recognizing, and separating multiple musical instruments from an audio clip. Jointist consists of an instrument recognition module that conditions the other two modules: a transcription module that outputs instrument-specific piano rolls, and a source separation module that utilizes instrument information and transcription results. The joint training of the transcription and source separation modules serves to improve the performance of both tasks. The instrument module is optional and can be directly controlled by human users. This makes Jointist a flexible user-controllable framework.

Our challenging problem formulation makes the model highly useful in the real world given that modern popular music typically consists of multiple instruments. Its novelty, however, necessitates a new perspective on how to evaluate such a model. In our experiments, we assess the proposed model from various aspects, providing a new evaluation perspective for multi-instrument transcription. Our subjective listening study shows that Jointist achieves state-of-the-art performance on popular music, outperforming existing multi-instrument transcription models such as MT3. We conducted experiments on several downstream tasks and found that the proposed method improved transcription by more than 1 percentage points (ppt.), source separation by 5 SDR, downbeat detection by 1.8 ppt., chord recognition by 1.4 ppt., and key estimation by 1.4 ppt., when utilizing transcription results obtained from Jointist.

## 1 INTRODUCTION

Transcription, or automatic music transcription (AMT), is a music analysis task that aims to represent audio recordings using symbolic notations such as scores or MIDI (Musical Instrument Digital Interface) files (Benetos et al., 2013; 2018; Piszczalski & Galler, 1977). AMT can play an important role in music information retrieval (MIR) systems since symbolic information – e.g., pitch, duration, and velocity of notes – determines a large part of our musical perception. A successful AMT should provide a denoised version of music in a musically-meaningful, symbolic format, which could ease the difficulty of many MIR tasks such as melody extraction (Ozcan et al., 2005), chord recognition (Wu & Li, 2018), beat tracking (Vogl et al., 2017), composer classification (Kong et al., 2020a), (Kim et al., 2020), and emotion classification (Chou et al., 2021). Finally, high-quality AMT systems can be used to build large-scale datasets as done by Kong et al. (2020b). This can, in turn, accelerate the development of neural network-based MIR systems as these are often trained using otherwise scarcely available audio aligned symbolic data (Brunner et al., 2018; Wu et al., 2020a; Hawthorne et al., 2018). Currently, the only available pop music dataset is the Slakh2100 dataset by Manilow et al. (2019). The lack of large-scale audio aligned symbolic dataset for pop music impedes the development of other MIR systems that are trained using symbolic music representations.

In the early research on AMTs, the problem is often defined narrowly as transcription of a single target instrument, typically piano (Klapuri & Eronen, 1998) or drums (Paulus & Klapuri, 2003), and whereby the input audio only includes that instrument. The limitation of this strong and then-unavoidable assumption is clear: the model would not work for modern pop music, which occupies a majority of the music that people listen to. In other words, to handle realistic use-cases of AMT, it

is necessary to develop a multi-instrument transcription system. Recent examples are Omnizart (Wu et al., 2021; 2020b) and MT3 (Gardner et al., 2021) which we will discuss in Section 2.

The progress towards multi-instrument transcription, however, has just begun. There are still several challenges related to the development and evaluation of such systems. In particular, the number of instruments in multiple-instrument audio recordings are not fixed. The number of instruments in a pop song may vary from a few to over ten. Therefore, it is limiting to have a model that transcribes a pre-defined fixed number of musical instruments in every music piece. Rather, a model that can adapt to a varying number of target instrument(s) would be more robust and useful. This indicates that we may need to consider instrument recognition and instrument-specific behavior during development as well as evaluation.

Motivated by the aforementioned recent trend and the existing issues, we propose Jointist – a framework that includes instrument recognition, source separation, as well as transcription. We adopt a joint training scheme to maximize the performance of both the transcription and source separation module. Our experiment results demonstrate the utility of transcription models as a pre-processing module of MIR systems. The result strengthens a perspective of transcription result as a symbolic representation, something distinguished from typical i) audio-based representation (spectrograms) or ii) high-level features (Choi et al., 2017; Castellon et al., 2021).

This paper is organized as follows. We first provide a brief overview of the background of automatic music transcription in Section 2. Then we introduce our framework, Jointist, in Section 3. We describe the experimental details in Section 4 and discuss the experimental results in Section 5. We also explore the potential applications of the piano rolls generated by Jointist for other MIR tasks in Section 6. Finally, we conclude the paper in Section 7.

## 2 BACKGROUND

While automatic music transcription (AMT) models for piano music are well developed and are able to achieve a high accuracy (Benetos et al., 2018; Sigtia et al., 2015; Kim & Bello, 2019; Kelz et al., 2019; Hawthorne et al., 2017; 2018; Kong et al., 2021), multi-instrument automatic music transcription (MIAMT) is relatively unexplored. MusicNet (Thickstun et al., 2016; 2017) and ReconVAT (Cheuk et al., 2021a) are MIAMT systems that transcribe musical instruments other than piano, but their output is a *flat* piano roll that includes notes from all the instruments in a single channel. In other words, they are not instrument-aware. Omnizart (Wu et al., 2019; 2021) is instrument-aware but it does not scale up well when the number of musical instruments increases as discussed in Section 5.3. MT3 Gardner et al. (2021) is currently the state-of-the-art MIAMT model. It formulates AMT as a sequence prediction task where the sequence consists of tokens representing musical notes. By adopting the structure of a natural language processing (NLP) model called T5 (Raffel et al., 2020), MT3 shows that a transformer-based sequence-to-sequence architecture can perform successful transcription by learning from multiple datasets for various instruments.

Although there have been attempts to jointly train a speech separation and recognition model (Shi et al., 2022), the joint training of transcription model together with a source separation model is still very limited in the MIR domain. For example, Jansson et al. (2019) use the F0 estimation to guide the singing voice separation. However, they only demonstrated their method with a monophonic singing track. In this paper, the Jointist framework extends this idea into polyphonic music. While Manilow et al. (2020) extends the joint transcription and source separation training into polyphonic music, their model is limited to up to five sources (Piano, Guitar, Bass, Drums, and Strings) which do not cover the diversity of real-world popular music. Our proposed method is trained and evaluated on 39 instruments, which is enough to handle real-world popular music. Hung et al. (2021) and Chen et al. (2021a) also use joint transcription and source separation training for a small number of instruments. However, they only use transcription as an auxiliary task during training while no transcription is performed during the inference phases. The model in Tanaka et al. (2020), on the other hand, is only capable of doing transcription. Their model applies a joint spectrogram and pitchgram clustering method to improve the multi-instrument transcription accuracy. A zero-shot transcription and separation model was proposed in Lin et al. (2021) but was only trained and evaluated on 13 classical instruments. Our proposed Jointist framework alleviates most of the problems mentioned above. Unlike existing models, Jointist is instrument-aware, and transcribes and separates only the instruments present in the input mix audio.

Figure 1: The proposed Jointist framework. This framework can transcribe/separate up to 39 different instruments as defined in Table 11 in the Appendix. $B$: batch size, $L$: audio length, $C$: instrument classes, $T$: number of time steps, $K$: number of predicted instruments. The dotted lines represent iterative operations that are repeated $K$ times. Best viewed in color.

## 3  JOINTIST

We designed the proposed framework, Jointist, to jointly train music transcription and source separation modules so as to improve the performance for both tasks. By incorporating an instrument recognition module, our framework is instrument-aware and compatible with up to 39 different instruments as shown in Table 11 of the Appendix.

As illustrated in Figure 1, Jointist consists of three modules: the instrument recognition module $f_{IR}$ (Figure 2), the transcription module $f_T$ (Figure 3), and the music source separation module $f_{MSS}$ (Figure 4). The $f_{IR}$ and $f_T$ share the same mel spectrogram $X_{mel}$ as the input, while $f_{MSS}$ uses the STFT spectrogram $X_{STFT}$ as the input.

Jointist is a flexible framework that can be trained end-to-end or individually. For the ease of evaluation, we trained $f_{IR}$ individually. Note that our framework could still work without $f_{IR}$. Since both $f_T$ and $f_{MSS}$ accept one-hot vectors $I_{cond}^i$ as the instrument condition, users could easily bypass $f_{IR}$ and manually specify the target instruments to be transcribed, making Jointist a human controllable framework. In Section 5.2-5.3, we show that joint training of $f_T$ and $f_{MSS}$ mutually benefits both modules. In the discussions below, we denote Jointist with $f_{IR}$ module as **end-to-end Jointist**; without $f_{IR}$ module as **controllable Jointist**.

### 3.1  INSTRUMENT RECOGNITION MODULE

Figure 2 in the Appendix shows the model architecture of the instrument recognition module $f_{IR}(X_{mel})$, which is inspired by the music tagging transformer proposed by Won et al. (2021a). Transformer networks (Vaswani et al., 2017) have been shown to work well for a wide range of MIR tasks (Chen et al., 2021b; Lu et al., 2021; Huang et al., 2018; Park et al., 2019; Won et al., 2019; Guo et al., 2022; Won et al., 2021a). We therefore designed our instrument recognition model to consist of a convolutional neural network (CNN) front-end and a transformer back-end. To prevent overfitting, we perform dropout after each convolutional block with a rate of $0.2$ (Srivastava et al., 2014). We conduct an ablation study as shown in Table 1 to find the optimal number of transformer layers, which turns out to be 4. The instrument recognition loss is defined as $L_{IR} = \text{BCE}(\hat{Y}_{cond}, Y_{cond})$ where BCE is the binary cross-entropy, $\hat{Y}_{cond}$ is a vector of the predicted instruments, and $Y_{cond}$ are the ground truth labels. The predicted $\hat{Y}_{cond}$ is then converted into $\hat{I}_{cond}^i$ to be used by $f_T$ and $f_{MSS}$ later.

### 3.2  TRANSCRIPTION MODULE

Figure 3 shows the model architecture of the transcription module, $f_T(X_{mel}, \hat{I}_{cond}^i)$ which is adopted from the onsets and frames model proposed by Hawthorne et al. (2017). This module is conditioned by the one-hot instrument vectors $\hat{I}_{cond}^i$ using FiLM (Perez et al., 2018), as proposed by Meseguer-Brocal & Peeters (2019) for source separation, which allows us to control which target instrument to transcribe. To ensure model stability, teacher-forced training is used, i.e. the ground truth $I_{cond}^i$ is used as the condition. During inference, we combine the onset rolls $\hat{Y}_{onset}$ and the frame rolls $\hat{Y}_{frame}$ based on the idea proposed by Kong et al. (2021) to produce the final transcription $\hat{Y}_{roll}$. In this procedure, $\hat{Y}_{onset}$ is used to filter out noisy $\hat{Y}_{frame}$, resulting in a cleaner $\hat{Y}_{roll}$. Note that $\hat{I}_{cond}^i$ could be

provided by either the $f_{\text{IR}}$ module or human users. The transcription loss is defined as $L_{\text{T}} = \sum_j L_j$, where $L_i = \text{BCE}(\hat{Y}_j, Y_j)$ is the binary cross entropy and $j \in \{\text{onset}, \text{frame}\}$.

## 3.3 Source Separation Module

Figure 4 shows the model architecture of the source separation module, $f_{\text{MSS}}(X_{\text{STFT}}, \hat{I}^i_{\text{cond}}, \hat{Y}^i_{\text{frame}})$ which is inspired by Jansson et al. (2017); Meseguer-Brocal & Peeters (2019). In addition to $\hat{I}^i_{\text{cond}}$ (either generated by $f_{\text{IR}}$ or provided by human users), our $f_{\text{MSS}}$ also uses $\hat{Y}^i_{\text{frame}}$ of the music. The source separation loss $L_{\text{MSS}} = \text{L2}(\hat{Y}^i_S, Y^i_S)$ is set as the L2 loss between the predicted source waveform $\hat{Y}^i_S$ and the ground truth source waveform $Y^i_S$. When combining the transcription output $\hat{Y}^i_{\text{frame}} = f_{\text{T}}(X_{\text{mel}}, \hat{I}^i_{\text{cond}})$ with $X_{\text{STFT}}$, we explored two different modes: summation $g(\hat{Y}^i_{\text{frame}}) + X_{\text{STFT}}$ and concatenation $g(\hat{Y}^i_{\text{frame}}) \oplus X_{\text{STFT}}$. Where $g$ is a linear layer that maps the 88 midi pitches $\hat{Y}^i_{\text{frame}}$ to the same dimensions as the $X_{\text{STFT}}$. An ablation study for the two different modes can be found in Figure 9 in the Appendix.

# 4 Experiments

## 4.1 Dataset

Jointist is trained using the Slakh2100 dataset (Manilow et al., 2019). This dataset is synthesized from part of the Lakh dataset (Raffel, 2016) by rendering MIDI files using a high-quality sample-based synthesizer with a sampling rate of 44.1 kHz. The training, validation, and test splits contain 1500, 225, and 375 pieces respectively. The total duration of the Slakh2100 dataset is 145 hours. The number of tracks per piece in the Slakh2100 dataset ranges from 4 to 48, with a median number of 9, making it suitable for multiple-instrument AMT. In the dataset, 167 different plugins are used to render the audio recordings.

Each plugin also has a MIDI number assigned to it, hence we can use this information to map 167 different plugins onto 39 different instruments as defined in Table 11 in the Appendix. We map the MIDI numbers according to their instrument types. For example, we map MIDI number 0-3 to piano, although they indicate different piano types. Finally, the original MIDI numbers only cover 0-127 channels. We add one extra channel (128) to represent drums so that our model can also transcribe this instrument.

We also conduct subjective evaluation to measure the perceptual quality of the transcription outputs, since it has been shown that higher objective metrics do not necessarily correlate to better perceptual quality (Simonetta et al., 2022; Luo et al., 2020). We selected 20 full-tracks of pop songs from 4 public datasets: Isophonics (Mauch et al., 2009), RWC-POP (Goto et al., 2002), JayChou29 (Deng & Kwok, 2017), and USPOP2002 (Berenzweig et al., 2004), considering the diversity of instrumentation and composition complexity. The metadata of the 20 songs and the participants are listed in Table 8 and Figure 5 of Appendix C.

## 4.2 Training

When training the modules of the Jointist framework, the audio recordings are first resampled to 16 kHz, which is high enough to capture the fundamental frequencies as well as some harmonics of the highest pitch $C_8$ on the piano (4,186 Hz) (Hawthorne et al., 2017; Kong et al., 2021). Following some conventions on input features (Hawthorne et al., 2017; Kong et al., 2021; Cheuk et al., 2021a), for the instrument recognition and transcription, we use log Mel spectrograms – with a window size of 2,048 samples, a hop size of 160 samples, and 229 Mel filter banks. For the source separation, we use STFT with a window size of 1,024, and a hop size of 160. This configuration leads to spectrograms with 100 frames per second. Due to memory constraints, we randomly sample a clip of 10 seconds of mixed audio to train our models. The Adam optimizer (Kingma & Ba, 2014) with a learning rate of 0.001 is used to train both $f_{\text{IR}}$ and $f_{\text{T}}$. For $f_{\text{MSS}}$, a learning rate of $1 \times 10^{-4}$ is chosen after preliminary experiments. All three modules are trained on two Tesla V100 32GB GPUs with a batch size of six each. PyTorch and Torchaudio (Paszke et al., 2019; Yang et al., 2022) are used to perform all of the experiments as well as the audio processing.

The overall training objective $L$ is a sum of the losses of the three modules, i.e., $L = L_{\text{IR}} + L_{\text{T}} + L_{\text{MSS}}$.

### 4.3 EVALUATION CONFIGURATION

#### 4.3.1 OBJECTIVE EVALUATION

A detailed explanation of each metric is available in Section B in the Appendix. The mAP score for $f_{\text{IR}}$ is calculated using the sigmoid output $\hat{Y}_{\text{cond}}$ in Figure 8. In practice, however, we need to apply a threshold to $\hat{Y}_{\text{cond}}$ to obtain the one-hot vector that can be used by $f_{\text{T}}$ and $f_{\text{MSS}}$ as the condition. Therefore, we also report the F1 score with a threshold of 0.5 in Table 1. We chose this threshold value to ensure the simplicity of the experiment, even though the threshold can be tuned for each instrument to further optimize the metric (Won et al., 2021b). To understand the model performance under different threshold values, we also report the mean average precision (mAP) in Table 1.

For $f_{\text{T}}$, we propose a new instrument-wise metric (Figure 9 and 10) to better capture the model performance for multi-instrument transcription. Existing literature uses mostly flat metrics or piece-wise evaluation (Hawthorne et al., 2017; Gardner et al., 2021; Kong et al., 2021; Cheuk et al., 2021a). Although this can provide a general idea of how good the transcription is, it does not show which musical instrument the model is particularly good or bad at. Since frame-wise metrics do not reflect the perceptual transcription accuracy (Cheuk et al., 2021b), we report only the note-wise (N.) and note-wise with offset (N&O) metrics in Table 2. We study the performance of both end-to-end and human controllable Jointist.

For $f_{\text{MSS}}$, we also report the instrument-wise metrics to better understand the model performance for each instrument (Figure 13 and Table 4).

#### 4.3.2 SUBJECTIVE EVALUATION

As mentioned in the Introduction, a successful AMT would provide a clear and musically-plausible symbolic representation of music. This suggests that synthesized audio from a transcribed MIDI file should resemble the original recording, given that the transcription was accurate. Therefore, participants were asked to use a DAW (Digital Audio Workstation), preferably GarageBand on Mac OS, to listen to the audio synthesized by the default software instruments and review the MIDI note display of each instrument present in the piece.

We focus on the comparison between Jointist and MT3 for the subjective evaluation. To generate the test MIDI files for the 20 selected songs, we used end-to-end Jointist and replicated the MT3 model following (Gardner et al., 2021)[1]. We propose four evaluation aspects (detailed explanation available in Appendix B.4): **instrument integrity**, **instrument-wise note continuity**, **overall note accuracy**, and **overall listening experience**. Participants imported the original audio of a piece in the DAW, alone with one of two anonymous MIDI transcriptions (by either Jointist or MT3), so that the signals are synchronized, allowing the participants to examine back and forth between the audio and MIDI. After that, they were asked to rate the two transcriptions according to the aspects mentioned above on a scale from 1 to 5 as explained in Appendix B.5. Participants were asked to pay more attention to *important instruments* such as drums, bass, guitar, and piano which plays a more important role in building the perceptions of genre, rhythm, harmony, and melody of a song.

## 5 RESULTS

### 5.1 INSTRUMENT RECOGNITION

Table 1 shows the mAP and F1 scores of the instrument recognition module $f_{\text{IR}}$ when a different number of transformer encoder layers are used. Both the mAP and F1 scores improve as the number of layers increases. The best mAP and F1 scores are reached when using four transformer encoder layers. Due to the instrument class imbalance in the Slakh2100 dataset, our $f_{\text{IR}}$ performance is relatively low for instrument classes with insufficient training samples such as clarinet, violin, or harp as shown in Figure 8 in Appendix E. Therefore, the weighted F1/mAP is higher than the

---

[1]https://github.com/magenta/mt3/blob/main/mt3/colab/music_transcription_with_transformers.ipynb

| #Layers (#Parameters) | mAP | | F1 | |
|---|---|---|---|---|
| | Macro | Weighted | Macro | Weighted |
| 1 (78.0M) | 71.8 | 91.6 | 61.5 | 85.4 |
| 2 (78.8M) | 72.1 | 92.1 | 65.0 | 86.2 |
| 3 (79.6M) | 73.7 | 92.2 | 63.7 | 86.9 |
| 4 (80.3M) | **77.4** | **92.6** | **70.3** | **87.6** |

Table 1: The accuracy of instrument recognition by the number of transformer layers.

| Model | Flat F1 | | Piece. F1 | | Inst. F1 | |
|---|---|---|---|---|---|---|
| | N. | N&O | N | N&O | N | N&O |
| Wu et al. (2019) | 26.6 | 13.4 | 11.5 | 6.30 | 4.30 | 1.90 |
| Gardner et al. (2021) | 76.0 | 57.0 | N.A. | N.A. | N.A. | N.A. |
| T | 57.9 | 25.1 | 59.7 | 27.2 | 48.7 | 23.2 |
| iT | 58.0 | 25.4 | N.A | N.A | N.A | N.A |
| pTS(s) | 58.4 | 26.2 | 61.2 | 28.5 | 50.6 | 24.7 |
| ipTS(s) | 58.4 | **26.5** | N.A | N.A | N.A | N.A |
| TS(s) | 47.9 | 18.6 | 45.7 | 18.4 | 34.9 | 15.8 |
| pTS(c) | **58.4** | 26.3 | **61.3** | **28.6** | **50.8** | **24.8** |
| TS(c) | 47.7 | 18.6 | 46.0 | 18.0 | 35.5 | 16.2 |

Table 2: Transcription accuracy for existing state-of-the-art models and our proposed Jointist. 'T' and 'S' specifies the trained $f_T$ and $f_{MSS}$ modules respectively. The prefix 'p-' represents that the transcription module is pretrained, 'i-' represents that $f_{IR}$ is used to obtain the instrument conditions, models without a prefix represent full control by human users in which their target instruments are exactly the ground truth instruments. (s) and (c) indicate whether the piano rolls are summed or concatenated, respectively, to the spectrograms.

macro F1/mAP. Nevertheless, we believe that our $f_{IR}$ is good enough to generate reliable instrument conditions for popular instruments such as drums, bass, and piano.

Note that human users can easily bypass this module by providing the one-hot instrument conditions directly to Jointist. Even if human users do not want to take over $f_{IR}$, Jointist could still work by activating all 39 instrument conditions. Therefore, $f_{IR}$ is an optional module of the framework and hence we keep its evaluation minimal.

## 5.2 Transcription

Table 2 shows the note-wise (N.) and note-wise with offset (N&O) transcription F1 scores for different models. In addition to training only the $f_T$ (T), we also explore the possibility of training both $f_T$ and $f_{MSS}$ jointly (TS) and study its effect on the transcription accuracy.

When evaluating the transcription module $f_T$, there will be cases where the $f_{IR}$ predicts instruments that are not in the mix which results in undefined F1 scores for that particular instrument. Given the fact that human users can bypass $f_{IR}$, we assume that the ground truth instrument labels $I_{cond}^i$ are the target instruments human users want to transcribe.

When training $f_T$ standalone for 1,000 epochs, we can achieve an instrument-wise F1 score of 23.2. We presumed that joint training of $f_T$ and $f_{MSS}$ would result in better performance as the modules would help each other. Surprisingly, the joint training of $f_T$ and $f_{MSS}$ from scratch results in a lower instrument-wise F1 score, 15.8. Only when we use a pretrained $f_T$ for 500 epochs and then continue training both $f_T$ and $f_{MSS}$ jointly, do we get a higher F1 score, 24.7. We hypothesize two reasons for this phenomenon. First, we believe that it is due to the noisy output $\hat{Y}_{frame}^i$ generated from $f_T$ in the early stage of joint training which confuses the $f_{MSS}$. The same pattern can be observed from the Source-to-Distortion Ratio (SDR) of the $f_{MSS}$ which will be discussed in Section 5.3. Second, multi-task training often results in a lower performance than training a model for a single task, partly due to the difficulty in balancing multiple objectives. Our model might be an example of such a case.

|  | Inst. Integrity | Note Continuity | Overall Note Acc. | Overall Lis. Exp. |
|---|---|---|---|---|
| MT3 | $2.66 \pm 1.04$ | $2.64 \pm 1.04$ | $2.76 \pm 1.05$ | $2.60 \pm 1.06$ |
| Jointist | $\mathbf{2.87} \pm 0.96$ | $\mathbf{2.92} \pm 1.03$ | $\mathbf{2.93} \pm 1.02$ | $\mathbf{2.91} \pm 1.03$ |
| $p$-value | 0.004 | 0.0001 | 0.02 | 2.7e-05 |

Table 3: Mean subjective scores (from 1 to 5) based on 20 participants, each rated 20 songs. The bottom row presents the $p$-values using T-test for the comparison between MT3 and Jointist.

When comparing the conditioning strategy, the difference between the summation and the concatenation modes is very subtle. The summation mode outperforms the concatenation mode by only 0.1 in terms of piece-wise F1 score; while the concatenation mode outperforms the summation mode by 0.001 in terms of flat F1 as well as the instrument-wise F1. We believe that the model has learned to utilize piano rolls to enhance the mel spectrograms. And therefore, summing up both the piano rolls after the linear projection with the mel spectrograms is enough to achieve this objective, and therefore no obvious improvement is observed when using the concatenation mode.

Compared to the existing methods, our model (24.8) outperforms Omnizart (Wu et al., 2019) (1.9) by a large margin in terms of Instrument-wise F1. The instrument-wise transcription analysis in Figure 9- 12 of Appendix E indicates that Jointist can handle heavily skewed data distributions (Figure 6) much better than Omnizart. While Omnizart only performs well for instruments with abundant labels available such as drums and bass, Jointist still maintains a competitive transcription accuracy as the instrument label availability decreases. This result proves that the proposed model with control mechanism (Jointist) is more scalable than the model (Omnizart) with a fixed number of output instruments.

Although our proposed framework (58.4) did not outperform the MT3 model (76.0) on the note-wise F1 score, Jointist outperforms MT3 in all aspects of our subjective evaluation (see Table 3). We notice that, while MT3 is able to capture clean and well defined notes in classical music, it struggles in popular music with complex timbre. Sometimes, the language model in MT3 dominates the acoustic model, producing a series of unexpected notes of transcription that are non-existent in the recording (examples can be found in our demo page[1]). MT3 is also less consistent in continuing the notes for an instrument, where a sequence of notes in the recording can jump back and forth between irrelevant instruments in the transcription, causing a notably lower perpetual score in "note continuity" in the subjective evaluation. Jointist is more robust than MT3 in these aspects. Moreover, Jointist is more flexible than MT3 in which human users can pick the target instruments that they are interested in. For example, if only a 'piano' condition is given, Jointist will only transcribe for piano. Whereas MT3 will simply transcribe all of the instrument found in the audio clip.

To test the robustness of proposed framework, we compare the flat transcription result between $f_{\text{IR}}$ generated $I_{\text{cond}}^{i}$ (prefix 'i-' in Table 2) and human specified conditions (assuming the target instruments are the instruments that users want to transcribe). We obtain a similar flat F1 score between the two approaches, which implies that $f_{\text{IR}}$ is good enough to pick up all of the necessary instruments for the transcription. Because false positive piano rolls and false negative piano rolls have undefined F1 scores, we are unable to report the piece-wise and instrument-wise F1 scores for $f_{\text{IR}}$ generated $I_{\text{cond}}^{i}$. The end-to-end transcription samples generated by Jointist are available online[2].

## 5.3 SOURCE SEPARATION

Table 4 shows that when $f_{\text{MSS}}$ (TS) is jointly trained with $f_{\text{T}}$, it outperforms a standalone trained $f_{\text{MSS}}$ (S). Similar to the discussion in Section 5.2, training TS from scratch does not yield the best result due to the noisy transcription output in the early stage which confuses the $f_{\text{MSS}}$ module. It can be seen that when using a pretrained $f_{\text{T}}$, $f_{\text{MSS}}$ is able to achieve a higher SDR (pTS). Our experimental results in both Section 5.2 and Section 5.3 show that the joint training of $f_{\text{T}}$ and $f_{\text{MSS}}$ helps both modules to escape local minima and therefore, achieve a better performance compared to training them independently. A full instrument-wise SDR analysis is available in Figure 13 of Appendix E, and the separated audio samples produced by Jointist are available online[1].

---

[2]https://jointist.github.io/Demo/

| Model | Instrument | Piece | Source |
|---|---|---|---|
| S | 1.52 | 3.24 | 3.03 |
| TS | 1.86 | 3.55 | 3.32 |
| pTS | **2.01** | **3.72** | **3.50** |
| Upper bound TS | 4.06 | 4.96 | 4.81 |

Table 4: Source-to-Distortion Ratio (SDR) for different models. T represents transcriptor, S represents separator. The 'p-' prefix represent pretrained model. The last row shows the upper bound when we use the ground truth piano rolls for source separation.

Spectrograms $X_{\text{STFT}}$ and posteriorgrams $\hat{Y}_{\text{frame}}^i$ can be combined using either "sum" (s) or "cat" (c). But there is only a minor difference in instrument/piece/source SDR between the former and latter (2.01/3.72/3.50 dB vs. 1.92/3.75/3.52 dB). Instead of $\hat{Y}_{\text{frame}}^i$, the binary piano rolls $\hat{Y}_{\text{roll}}^i$ can also be used as the condition for our $f_{\text{MSS}}$, however, this does not yield a better result (Table 9 in Appendix F). We also explore the upper bound of source separation performance when the ground truth $Y_{\text{T}}^i$ is used (last row of Table 4). When $f_{\text{MSS}}$ has access to accurate transcription results, its SDR can be greatly improved. This shows that AMT is an important MIR task that could potentially benefit other downstream tasks such as music source separation. In the next Section, we will explore applications of the transcription output produced by Jointist.

## 6 Applications of Jointist

### 6.1 Downbeat, Chord, and Key Estimations

It is intuitive to believe that symbolic information is helpful for beat and downbeat tracking, chord estimation, and key estimation. Literature has indicated that the timing of notes is highly related to beats (Matthew E. P. Davies, 2021). Downbeats, on the other hand, correspond to bar boundaries and are often accompanied by harmonic changes (Durand et al., 2016). The pitch information included in piano rolls offers explicit information about the musical key and chords (Pauws, 2004; Humphrey & Bello, 2015). It thus also provides strong cues for modeling downbeats since they are correlated with harmonic changes. Given these insights, we attempt to apply our proposed hybrid representation to improve the performance of these tasks. Detail descriptions of the experimental setup and datasets used are available in Appendix G.

Table 5 presents the evaluation results for each task. We report the frame-wise Major/Minor score for chord, F1 score for downbeat tracking, and the song-level accuracy. We observe that the model with the hybrid representation can consistently outperform the one that uses only spectrograms, and this across all three tasks. This can be attributed to the advantage of piano rolls that provide explicit rhythmic and harmonic information to SpecTNT, which is frequency-aware (i.e., not shift invariant along the frequency axis).

### 6.2 Music Classification

We experimented to see if the piano rolls produced by Jointist are also useful for music classification. The MagnaTagATune dataset (Law et al., 2009) is a widely used benchmark in automatic music tagging research. We used ≈21k tracks with top 50 tags following previous work by Won et al. (2020b).

| Eval Data | Input | Downbeat (F1) | Chord (MajMin) | Key (Acc) |
|---|---|---|---|---|
| Isophonics | Audio Only | 72.8 | 79.8 | 72.8 |
| | Hybrid | **74.6** | **81.2** | **75.2** |
| RWC-POP | Audio Only | 62.0 | 76.6 | - |
| | Hybrid | **66.3** | **78.5** | - |

Table 5: Comparison of with and without pianoroll representation on each task on two datasets.

| Input | Model | ROC-AUC | PR-AUC |
|---|---|---|---|
| Audio Only | Pons & Serra (2019) | 91.06 | 44.93 |
| | Lee et al. (2017) | 90.58 | 44.22 |
| | Won et al. (2020b) | 91.29 | 46.14 |
| | Won et al. (2020a) | 91.27 | 46.11 |
| Piano Roll Only | Transformer | 89.38 | 40.63 |
| Hybrid | Transformer | 90.90 | 44.00 |

Table 6: Music tagging results on MagnaTagATune Dataset. The audio-only baseline systems are MusiCNN Pons & Serra (2019), Sample-level CNN Lee et al. (2017), Short-chunk ResNet Won et al. (2020b), and Harmonic CNN Won et al. (2020a).

Since music classification using symbolic data is a less explored area, we design a new architecture that is based on the music tagging transformer (Won et al., 2021a). The size of our piano roll input is (B, 39, 2913, 88), where B is the batch size, 39 is the number of instruments, 2913 is the number of time steps, and 88 is the number of MIDI note bins. The CNN front-end has 3 convolutional blocks with residual connections, and the back-end transformer is identical to the music tagging transformer (Won et al., 2021a). The area under the receiver operating characteristic curve (ROC-AUC) and the area under the precision-recall curve (PR-AUC) are reported as evaluation metrics.

As shown in Table 6, the hybrid model that uses both audio and piano roll features does not outperform the audio-only model. We thus further experimented to see whether the piano roll as the only input is enough for music classification. Surprisingly, using piano roll-only yields competitive results. This opens up new possibilities to apply music classification to pure symbolic music datasets (or audio datasets converted to symbolic using frameworks like Jointist). Introducing the symbolic features as the extra modality for classifier models also allows us to discover new aspects. We observed the following phenomena during our experiments:

- The model performance for piano roll-only is comparable to audio-only methods when an instrument tag is one of the 39 instruments for which Jointist was explicitly trained (e.g., cello, violin, sitar), or a genre is highly correlated with the Slakh2100 dataset (e.g., rock, techno).

- The piano roll-only model performance drops when a tag is not related to the 39 instruments (e.g., female vocal, male vocal; compared to guitar, piano), or when the tag is related to acoustic characteristics that cannot be captured by piano rolls (e.g., quiet)

While these conclusions are interesting, this is exploratory work and an in-depth exploration is left as future work.

## 7 CONCLUSION

In this paper, we introduced Jointist, a flexible framework for instrument recognition, transcription, and source separation. In this framework, human users can easily provide the target instruments that they are interested in, such that Jointist transcribes and separates only these target instruments Joint training is used in Jointist to mutually benefit both transcription and source separation performance. In our extensive experiments, we show that Jointist outperforms existing multi-instrument automatic music transcription models. In our listening study, we confirm that the transcription result produced by Jointist is on par with state-of-the-art models as such MT3 (Hawthorne et al., 2021). We then explored different practical use cases of our framework and show that transcriptions resulting from our proposed framework can be useful to improve model performance for tasks such as downbeat tracking, chord, and key estimation.

In the future, Jointist may be further improved, for instance, by replacing the ConvRNN with Transformers as done in Hawthorne et al. (2021). From our experiments, the power of using symbolic representations for tasks such as tagging was shown, hence we hope to see more attempts to use symbolic representations, complementary to or even replace audio representations, to progress towards a more complete, multi-task music analysis system.

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

## A    MODEL ARCHITECTURES

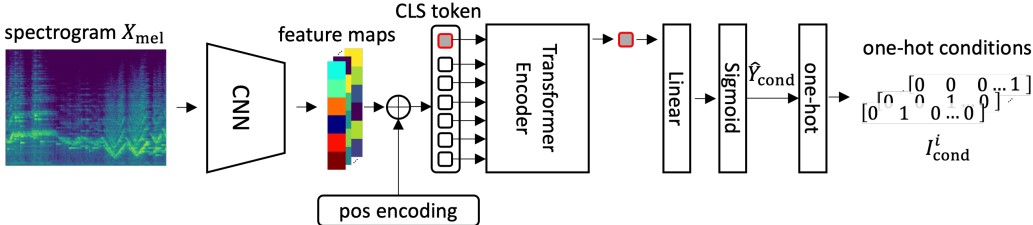

Figure 2: Model architecture for the $f_{\text{IR}}$ module which consists of a CNN backbone and a transformer encoder. The classification (CLS) token in the transformer encoder is used to extract the musical instruments present in the spectrogram. The CNN front end has six convolutional blocks with output channels $[64, 128, 256, 512, 1024, 2048]$. Each convolutional block has two convolutional layers with kernel size (3,3), stride (1,1), and padding (1,1), followed by average pooling (2,2). Sigmoid activation is used to convert the output into probability. Finally, a threshold ($p > 0.5$) is applied to the sigmoid output and obtain the one-hot vectors as the instrument conditions.

Note that due to the one-hot nature of the instrument condition, human users can easily take over $f_{\text{IR}}$ and specify the target instruments they are interested. Otherwise, all 39 instruments conditions could be used to perform a full-fledge transcription, or make sure of $f_{\text{IR}}$ to automatically determine the instruments present in the audio. This makes Jointist a flexible framework.

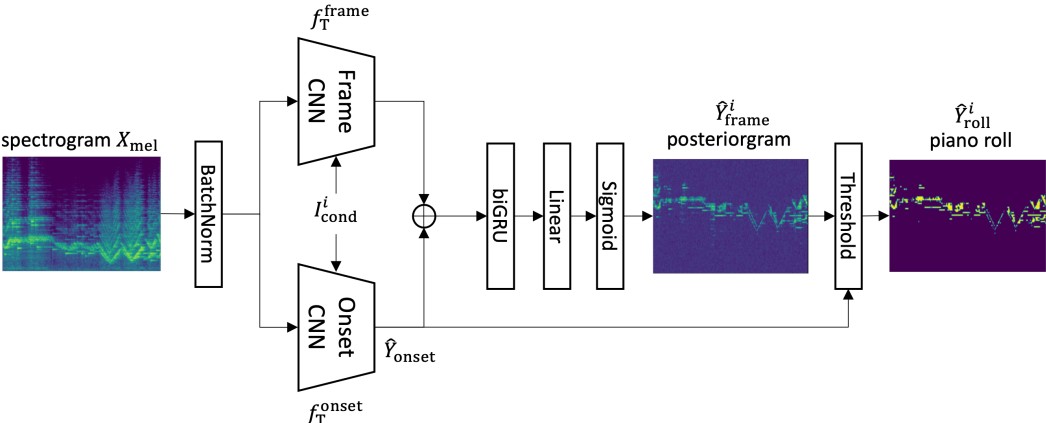

Figure 3: Model architecture for the $f_{\text{T}}$ module. It consists of a batch normalization layer (Ioffe & Szegedy, 2015), a frame model $f_{\text{T}}^{\text{frame}}$, and an onset model $f_{\text{T}}^{\text{onset}}$. Both $f_{\text{T}}^{\text{frame}}$ and $f_{\text{T}}^{\text{onset}}$ follow the same architecture which consist of four conditional convolutional blocks that takes in $\hat{I}_{\text{cond}}^i$ via a FiLM layer, followed by a linear layer with ReLU activation, one layer of bidirectional GRU (biGRU) (Chung et al., 2014) of 256 hidden dimension, and a linear layer with sigmoid activation which projects the hidden dimension into 88 notes. Each conditional convolutional block contains two 2D convolutional layers with batch normalization and ReLU activation followed by a average pooling size of $(2, 2)$.

The outputs from $f_{\text{T}}^{\text{frame}}$ and $f_{\text{T}}^{\text{onset}}$ are concatenated together, resulting in a tensor of 172 dimension. Then this tensor is passed to another biGRU layer with 256 hidden dimensions followed by a linear layer with sigmoid activate which projects the hidden features back into 88 as the posteriorgram $\hat{Y}_{\text{frame}}^i$.

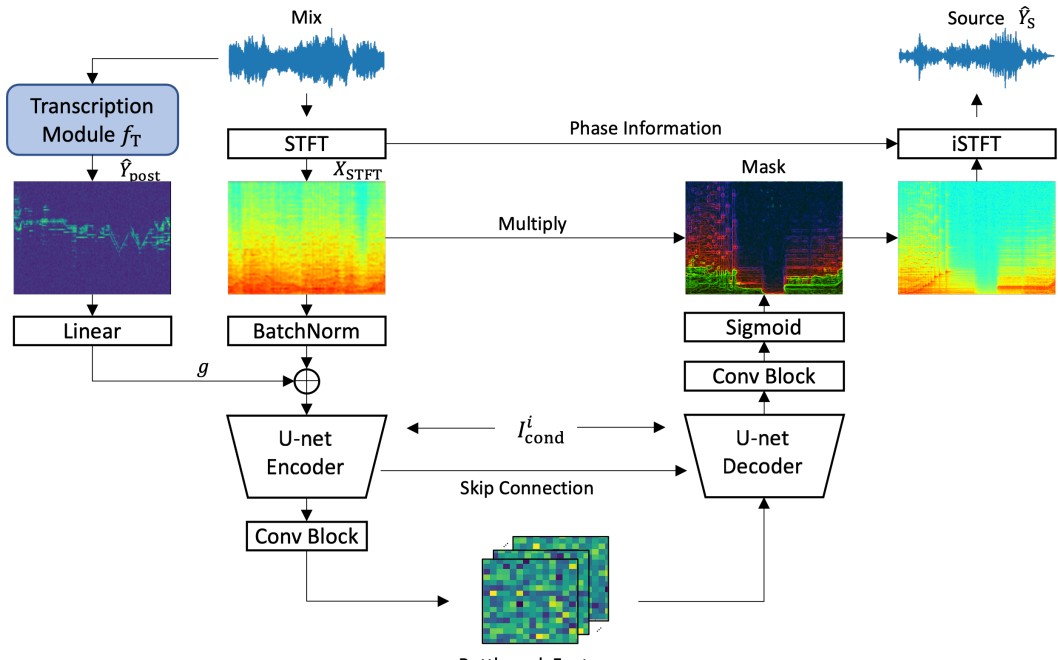

Figure 4: Model architecture for the $f_{\mathrm{MSS}}$ module has a U-net as the main component which takes $X_{\mathrm{STFT}}$ as input. FiLM is added to the U-net such that the output source $\hat{Y}_S^i$ can be controlled by $\hat{I}_{\mathrm{cond}}^i$. A linear layer is used to project $\hat{Y}_{\mathrm{frame}}^i = f_{\mathrm{T}}(X_{\mathrm{mel}}, \hat{I}_{\mathrm{cond}}^i)$ into a tensor $g$ with the same shape as $X_{\mathrm{STFT}}$. Then the U-net encoder takes in the merged feature of the batch normalized $X_{\mathrm{STFT}}$ and $g$ as the input. Different merging methods such as summation and concatenation as well as spectrogram patch has been studied in Table 9. In the spectrogram patch approach, we added (sum) our posteriorgram feature $g$ directly into

The U-net encoder consists of six convolution blocks with output channels [32, 64, 128, 256, 384, 384]. Each convolution block contains two bias-less convolutional layers with kernel size (3,3), stride size (1,1), padding (1,1) and average pooling size (2,2) followed by a batch normalization layer. The output of the convolutional layers are combined with FiLM condition as in Meseguer-Brocal & Peeters (2019). There is an extra convolution block after the encoder which has both input and output channel equal to 384, kernel size of (2,2) and padding of (1,1).

The U-net decoder follows a similar architecture to the U-net encoder to reverse the process. It consists of six Transposed convolution blocks with output channels [384, 384, 256, 128, 64, 32]. Each convolution block contains one bias-less transposed convolutional layers with kernel size (3,3), stride size (1,1), padding (0,0) followed by a batch normalization layer. Similarly, there is a convolution block after the decoder which has both input and output channel equal to 32, kernel size of (1,1) and padding of (0,0).

The output of the U-net Decoder is a spectrogram mask with values between [0,1]. The original magnitude STFT (mix) is multiplied with this mask to produce the separated magnitude STFT (source). To invert it back to the waveform, we use inverse STFT which requires also the phase information. Since our separated magnitude STFT has no phase, we use the original phase information in the mix STFT.

We also conducted an ablation on using the continuous value posteriorgrams $\hat{Y}_{\mathrm{frame}}^i \in [0, 1]$ versus the binary value piano rolls $\hat{Y}_{\mathrm{roll}}^i \in \{0, 1\}$ in Table 9, and we found that using posteriorgrams is generally better than piano rolls.

# B   EVALUATION METRICS

## B.1   MEAN AVERAGE PRECISION

We first calculate the instrument-wise average precision ($AP_s^i$) for each audio sample $s$ using scikit-learn package (Pedregosa et al., 2011) which is defined as

$$\text{AP}_s^i = \sum_k (R_k^i - R_{k-1}^i)P_k^i, \tag{1}$$

where where $P_k^i$ and $R_k^i$ are the precision and recall at the $k$-th threshold for the instrument $i$ defined in Table 11.

Then the mean average precision ($mAP^i$) for instrument $i$ across the test set with size $N$ can be calculated using

$$\text{mAP}^i = \frac{1}{N}\sum_{s=1}^{N}\text{AP}_s^i \tag{2}$$

This metric measures the instrument recognition performance across different threshold values, and hence it is not affected by an incorrectly selected threshold value. Due to the skewed distribution of different instrument labels (as shown in Figure 6), we report both the **macro** and **weighted** mAP. The **macro mAP** calculate the average mAP score across different instrument unweighted as

$$\text{macro mAP} = \frac{\sum_{i=1}^{I}\text{mAP}^i}{I}, \tag{3}$$

where $I$ is the total number of instruments which is 39 according to Table 11. Macro mAP is susceptible to instruments such as Bassoon, Cello, and Recorder with only a few labels and hence it is not a reliable metrics when the data distribution is heavily skewed. But it gives us an idea on whether our model is doing well across all instrument classes. A low macro score implies that our model performance poorly on a specific instrument classes.

To account for the skewed data distribution, we also report **weighted mAP** which is calculated as

$$\text{weighted mAP} = \frac{\sum_{i=1}^{I}(\text{mAP}^i \cdot w(i))}{I}, \tag{4}$$

where $w(i)$ is the weighting for the $i$-th instrument proportional to its label count in the test set. This metric is more reliable than the macro metric, since it weights more towards classes with more labels and less towards classes with less labels. In other words, one prediction mistake in instrument classes with few labels would not seriously affect the weighted mAP score.

With both metrics reported together with Figure 8, we understand that our model performance is highly correlated to the label availability as shown in Figure 6.

## B.2   NOTE-WISE TRANSCRIPTION F1 SCORE

We use the standard `transcription.precision_recall_f1_overlap` function from `mir_eval` to calculate the note-wise F1 score (notated as $F_s^i$) for each instrument $i$ present in each audio sample $s$. We follow the standard note-wise F1 setting where both the predicted pitch needs to be same as the ground truth label and the predicted onset needs to be within 50ms of the ground truth onset in order to be considered as a correct transcription.

Our **piece-wise note-wise F1** score is computed as

$$\text{F1}_{\text{piece}} = \frac{1}{S}\sum_s \frac{1}{I_s}\sum_i F_s^i, \tag{5}$$

where $I_s$ is a variable representing the number of ground truth instruments present in the audio sample $s$, and $S$ is a constant representing the total number of audio samples in the test set. While

this metrics reflect how well our model performance in general, it provides no information on which instrument our model is particularly good at or bad at.

To measure the performance on different instruments we proposed the **instrument-wise note-wise F1 score** which is defined as

$$\text{F1}_{\text{instrument}} = \frac{1}{I} \sum_i \frac{1}{S_i} \sum_s F_s^i,$$
(6)

where $S_i$ is a variable representing the number of audio samples that contains instrument $i$, and $I = 39$ is a constant representing the number of musical instruments supported by our model as listed in Table 11. Note that although instrument-wise evaluation was mentioned in (Cheng et al., 2013), they have always 10 instruments in each audio sample which makes their evaluation method much simpler than ours. In our case, we have varying number of instruments per samples. Hence, the instrument-wise evaluation for Slakh2100 is not as straight forward as (Cheng et al., 2013).

### B.3 SOURCE-TO-DISTORTION RATIO (SDR)

We followed the standard SDR defined in (Vincent et al., 2006),

$$\text{SDR} = 10 \log_{10} \left( \frac{\|s_{\text{target}}\|^2}{\|s_{\text{target}} - \hat{s}\|^2} \right),$$
(7)

where $s_{\text{target}}$ is the ground truth waveform and $\hat{s}$ is the predicted waveform. Since we have different number of instruments $I_s$ per audio sample $s$ in Slakh2100, we need to define new metrics to handle this case. For the sake of disccusion below, we define $\text{SDR}_s^i$ as the SDR for instrument $i$ of sample $s$.

**source-wise SDR** is defined as the mean SDR for all sources $N_{\text{all}}$ present in the dataset,

$$\text{SDR}_{\text{source}} = \frac{1}{N_{\text{all}}} \sum_s \sum_i \text{SDR}_s^i.$$
(8)

This metric is the most straight forward yet the least informative one.

To understand how well our model performs for each audio sample (music piece), we define **piece-wise SDR** as follows

$$\text{SDR}_{\text{piece}} = \frac{1}{N_S} \sum_s \frac{1}{N_s^I} \sum_i \text{SDR}_s^i,$$
(9)

where $N_s^I$ is the number of instruments for audio sample (music piece) $s$ and $N_S$ is the total number of audio samples (music pieces) in the test set. This metric allows us to understand the model performance for each piece.

Similarly, to understand the model performance for each instrument, **instrument-wise SDR** is used. It is defined as

$$\text{SDR}_{\text{instrument}} = \frac{1}{I} \sum_i \frac{1}{N_S^i} \sum_s \text{SDR}_s^i,$$
(10)

where $N_S^i$ is the number of pieces containing instrument $i$ and $I = 39$ as in Equation 6.

### B.4 SUBJECTIVE METRICS

- **Instrument integrity**: How accurate is the recognition of important instruments? If an important instrument is missed or falsely included as a dominant instrument, it is regarded as poor in instrument integrity.

- **Instrument-wise note continuity**: How is the note continuity of each important instrument? For instance, if a sequence of piano notes are assigned by the system to guitar, organ, and piano from segment to segment, it is regarded as poor note continuity for piano.
- **Overall note accuracy**: How accurate are the notes of important instruments when listening to them as a whole? Do they miss important notes or insert false notes?
- **Overall listening experience**: What is the overall quality of the transcription?

### B.5 RATING CRITERIA FOR SUBJECTIVE EVALUATION OF TRANSCRIPTION

Table 7 list the score criteria for the four aspects proposed in Section 4.3.2.

| Score | Criterion for Instrument Integrity |
|---|---|
| 5 | Perfect. |
| 4 | Some unimportant instruments are missed or falsely recognized, and they do not affect the overall listening experience. |
| 3 | Some important instruments are missed or falsely recognized, and they moderately affect the overall listening experience. |
| 2 | More important instruments are missed or falsely recognized, and they obviously degrade the overall listening experience. |
| 1 | Instruments recognized are completely wrong. |

| Score | Criterion for Instrument-wise Note Continuity |
|---|---|
| 5 | Perfect. |
| 4 | Some unimportant instruments' notes are less continuous, and they do not affect the overall listening experience. |
| 3 | Some important instruments' notes are not continuous, and they moderately affect the overall listening experience. |
| 2 | More important instruments' notes are not continuous, and they obviously degrade the overall listening experience. |
| 1 | The notes are assigned to instruments at random, where no rule can be concluded. |

| Score | Criterion for Overall Note Accuracy |
|---|---|
| 5 | Perfect. |
| 4 | Some unimportant instruments' notes are less accurate, and they do not affect the overall listening experience. |
| 3 | Some important instruments' notes are inaccurate, and they moderately affect the overall listening experience. |
| 2 | More important instruments' notes are inaccurate, and they obviously degrade the overall listening experience. |
| 1 | The notes are completely messy, and one cannot identify the song from the notes. |

| Score | Criterion for Overall Listening Experience |
|---|---|
| 5 | Excellent |
| 4 | Good |
| 3 | Fair |
| 2 | Poor |
| 1 | Awful |

Table 7: Rating criteria for the subjective transcription quality.

## C METADATA FOR SUBJECTIVE EVALUATION

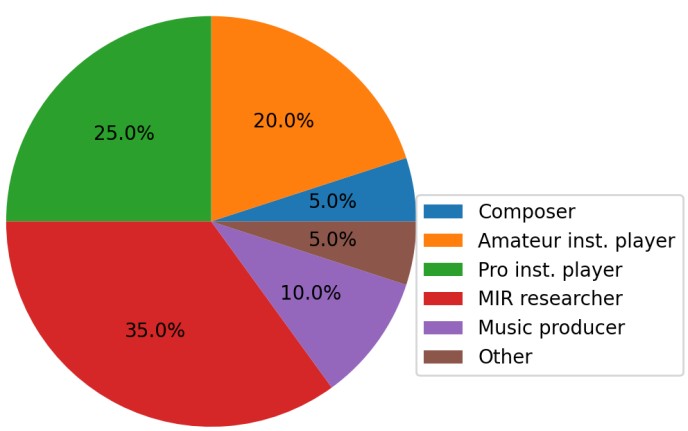

Figure 5: Music background for the subjective evaluation participants.

| Artist | Song Title (or ID) |
|---|---|
| **Isophonics** | |
| Beatles | Let It Be |
| Beatles | Lucy In The Sky With Diamonds |
| Beatles | Yellow Submarine |
| Michael Jackson | Black or White |
| Michael Jackson | I Want You Back |
| Michael Jackson | Off the Wall |
| Queen | Bohemian Rhapsody |
| Queen | I Want to Break Free |
| **RWC-POP** | |
| RWC | RM-P004 |
| RWC | RM-P033 |
| RWC | RM-P047 |
| RWC | RM-P083 |
| RWC | RM-P096 |
| **JayChou29** | |
| Jay Chou | chao-ren-bu-hui-fei |
| Jay Chou | gei-wo-yi-shou-ge-de-shi-jian |
| Jay Chou | ju-hua-tai |
| **USPOP2002** | |
| PapaRoach | Last Resort |
| Radiohead | Karma Police |
| Ricky Martin | Livin La Vida Loca |
| Spice Girls | Become One |

Table 8: Metadata of the selected 20 full-tracks for subjective evaluation. The audio of Isophonics, JayChou29, and USPOP2002 may be available at `https://github.com/krist311/chords-recognition`.

## D DATA DISTRIBUTION

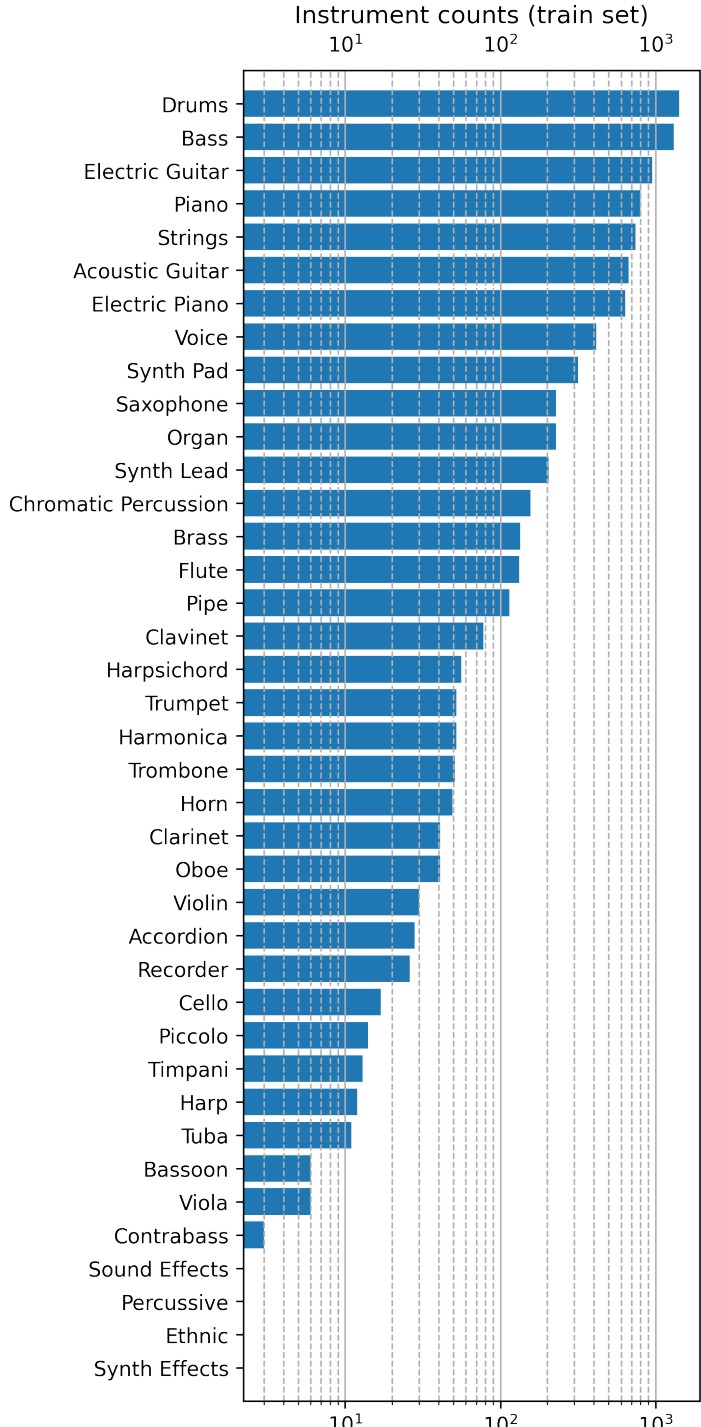

Figure 6: Instrument label count for the train set of Slakh2100. The label count is in log scale.

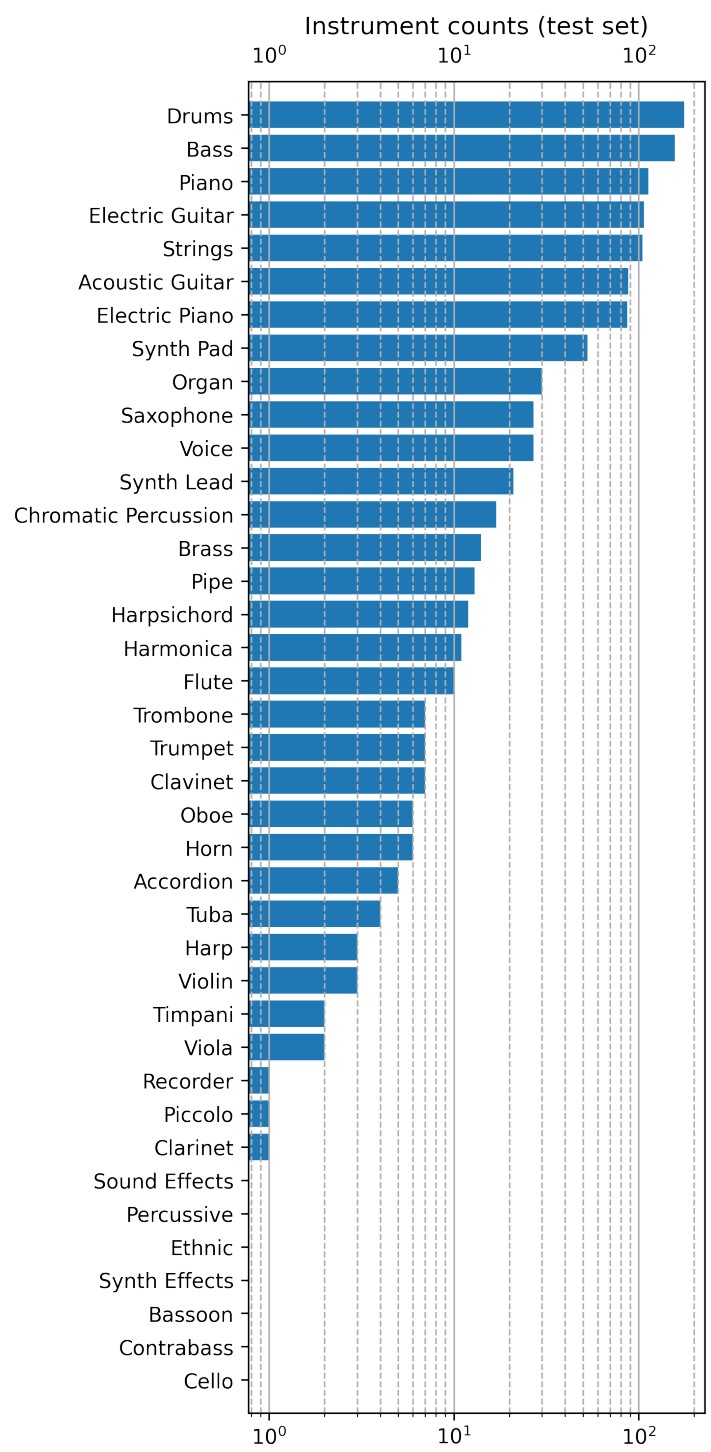

Figure 7: Instrument label count for the test set of Slakh2100. The label count is in log scale.

# E  INSTRUMENT-WISE PERFORMANCE

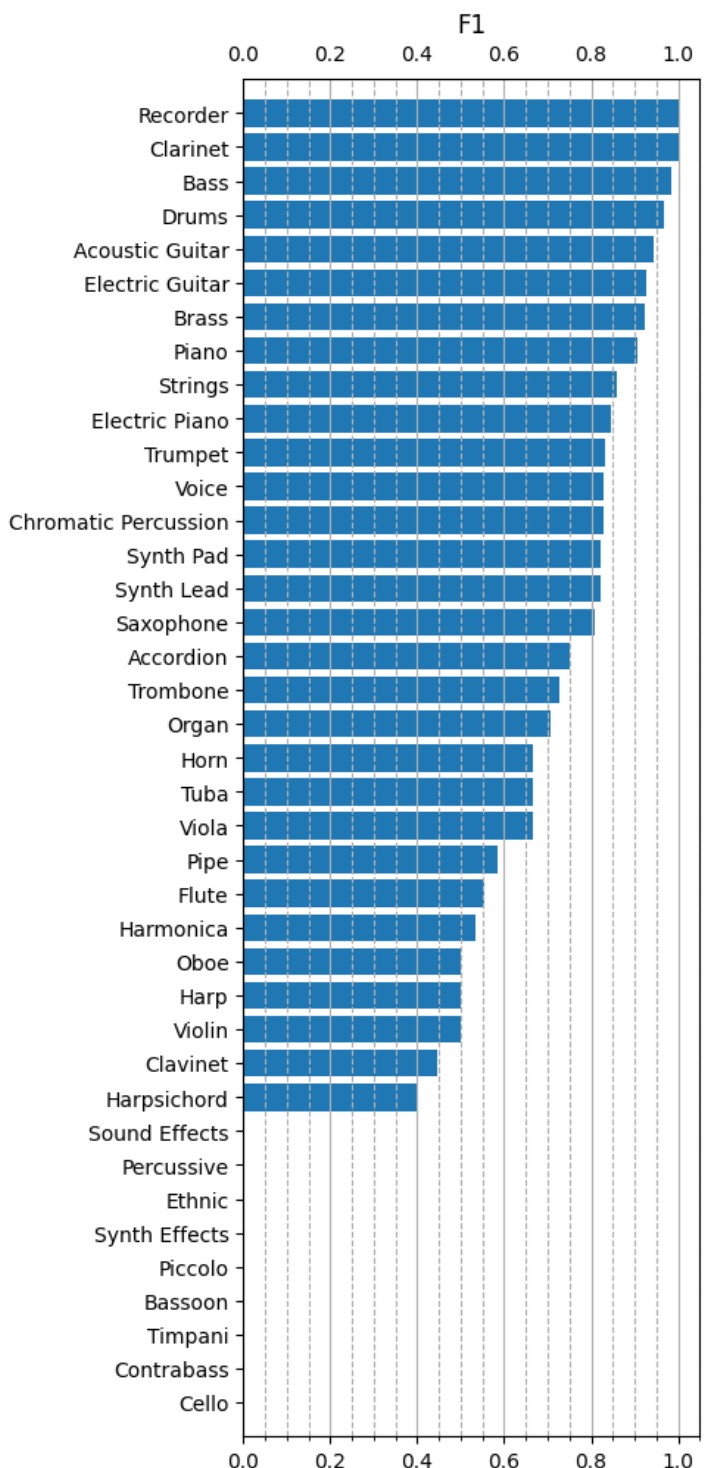

Figure 8: Instrument-wise F1 score for the $f_{\text{IR}}$ module in the test set of Slakh2100.

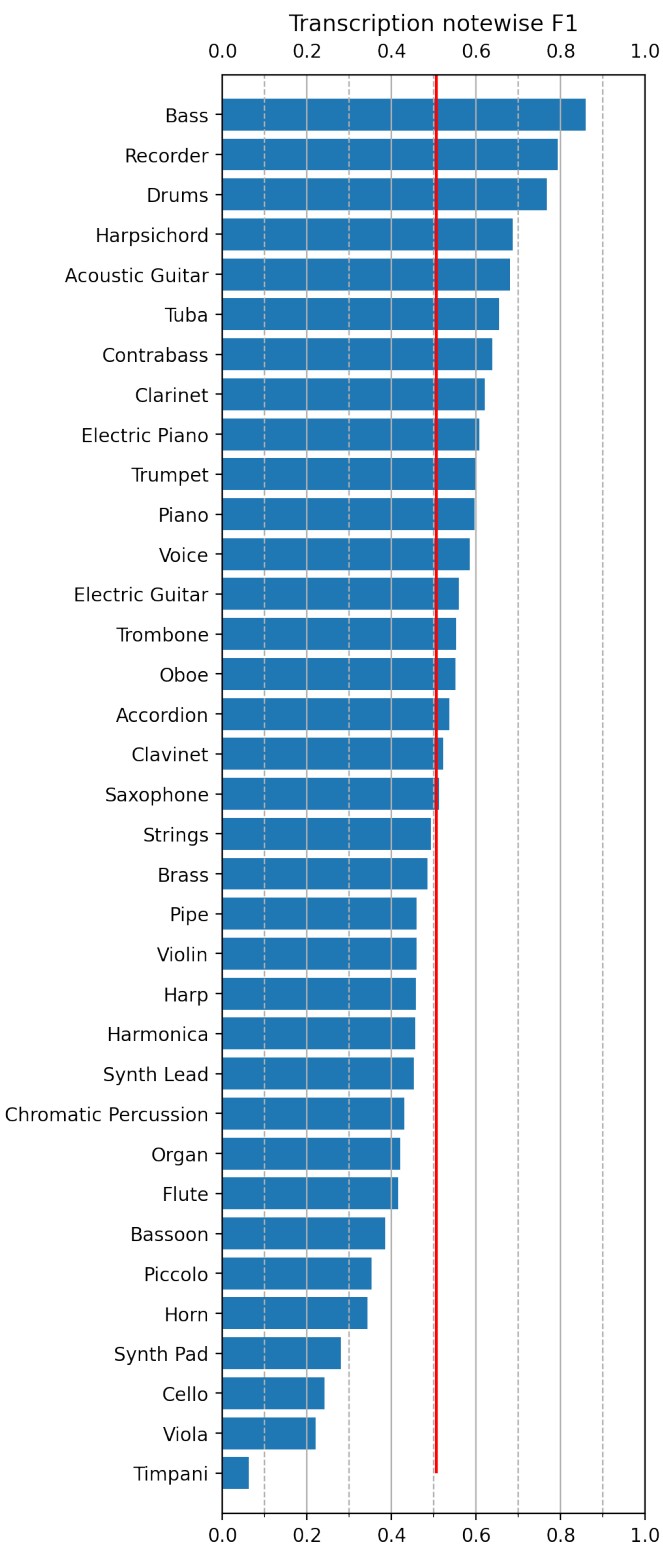

Figure 9: Instrument-wise note F1 score for the $f_\mathrm{T}$ module in the test set of Slakh2100. The red line represents the average F1 score.

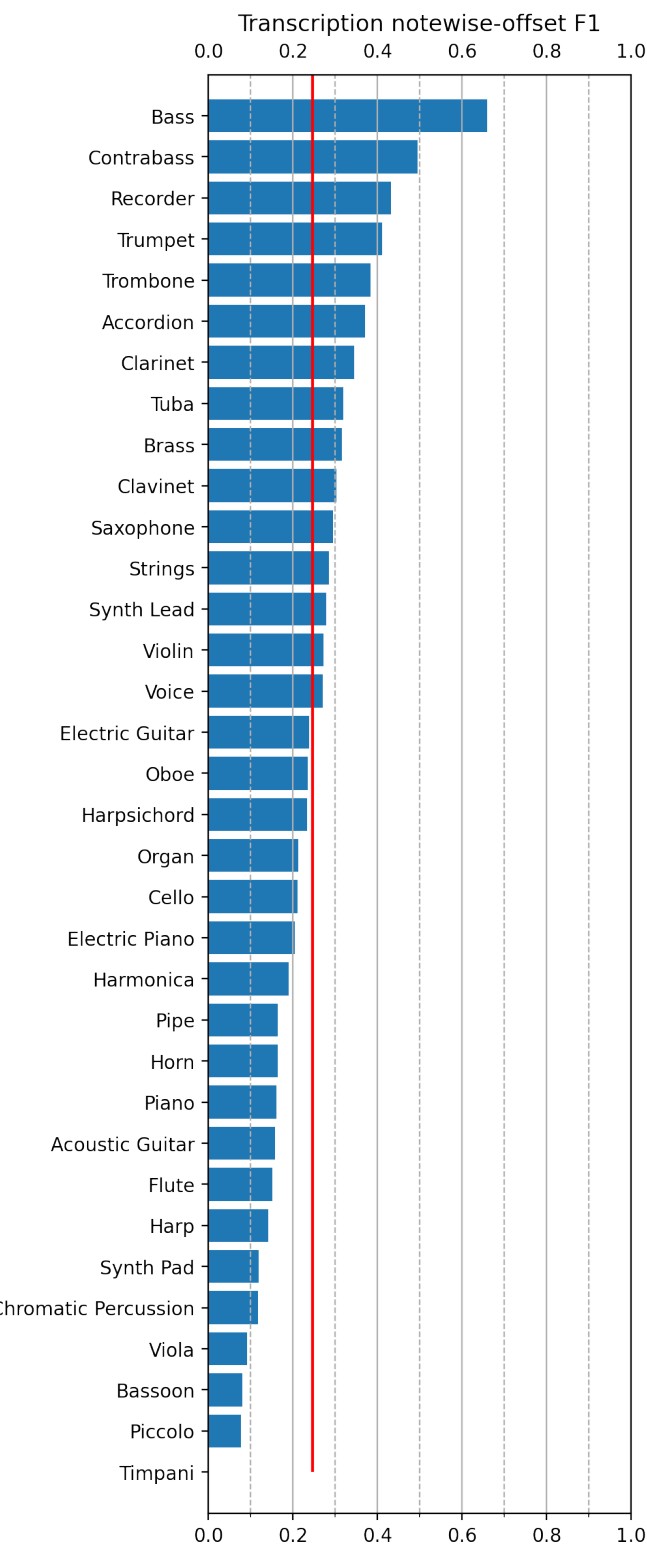

Figure 10: Instrument-wise note with offset F1 score for the $f_T$ module in the test set of Slakh2100. The red line represents the average F1 score.

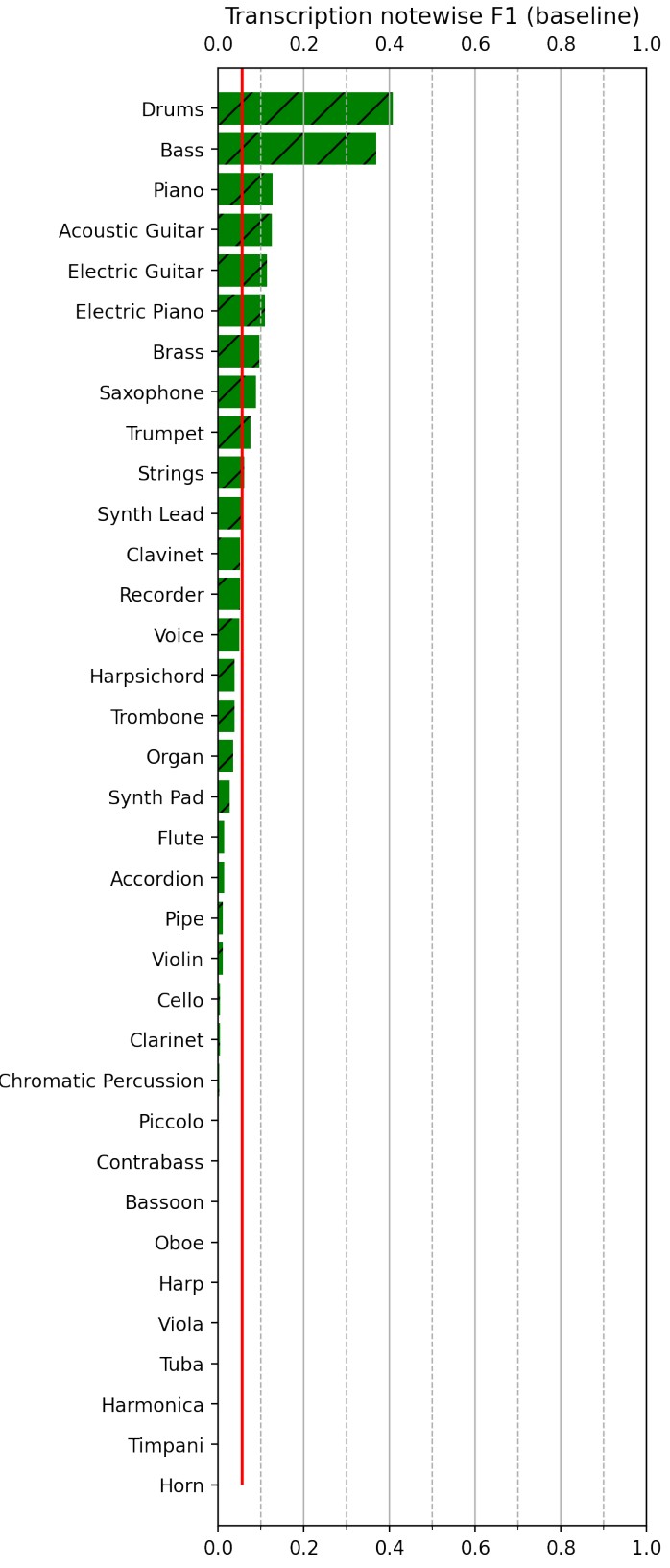

Figure 11: Instrument-wise note F1 scores (with and without offset) for the baseline model Wu et al. (2021) in the test set of Slakh2100. The red line represents the average F1 score.

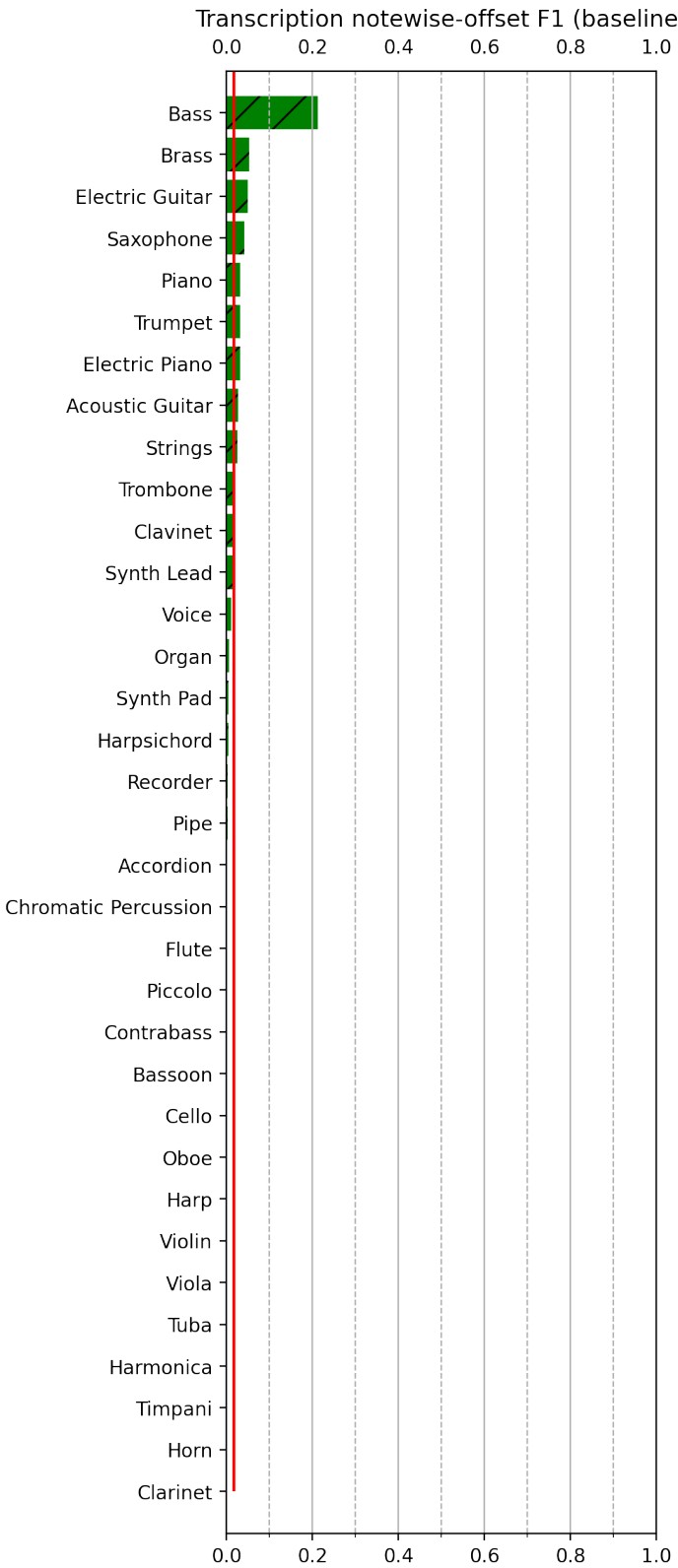

Figure 12: Instrument-wise note F1 scores (with and without offset) for the baseline model Wu et al. (2021) in the test set of Slakh2100. The red line represents the average F1 score.

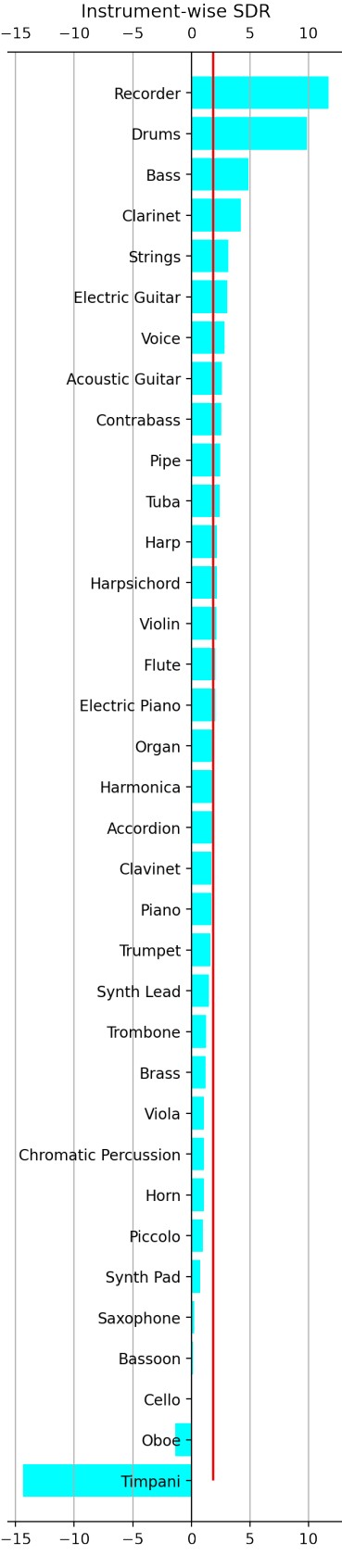

Figure 13: Instrument-wise source-to-distortion ratio (SDR) F1 score for the $f_T$ module in the test set of Slakh2100. The red line represents the average F1 score.

# F   ABLATION STUDY

| feature merge | Model | Full length SDR | | | 10s SDR | | |
|---|---|---|---|---|---|---|---|
| | | inst | piece | source | inst | piece | source |
| sum | TS (posterior) | 1.86 | 3.55 | 3.32 | 3.47 | 5.60 | 4.54 |
| | pTS (posterior) | **2.01** | 3.72 | 3.50 | **3.69** | 5.86 | 4.81 |
| | TS+STE (binary) | 1.30 | 3.30 | 3.06 | 2.40 | 5.13 | 4.16 |
| | pTS+STE (binary) | 1.67 | 3.30 | 3.06 | 3.29 | 5.36 | 4.41 |
| concat | TS (posterior) | 1.80 | 3.53 | 3.31 | 3.21 | 5.50 | 4.51 |
| | pTS (posterior) | 1.92 | **3.75** | **3.52** | 3.50 | **6.03** | **4.90** |
| | TS+STE (binary) | 1.89 | 3.60 | 3.37 | 3.63 | 5.70 | 4.66 |
| | pTS+STE (binary) | 1.72 | 3.47 | 3.22 | 3.18 | 5.78 | 4.61 |
| spec patch | TS (posterior) | 1.79 | 3.46 | 3.24 | 3.35 | 5.30 | 4.43 |
| | pTS (posterior) | 1.91 | 3.61 | 3.40 | 3.58 | 5.52 | 4.65 |
| | TS+STE (binary) | 1.74 | 3.46 | 3.24 | 3.50 | 5.28 | 4.42 |
| | pTS+STE (binary) | 1.71 | 3.47 | 3.23 | 3.28 | 5.40 | 4.53 |

Table 9: Ablation study for the source-to-distortion ratio (SDR) achieved by $f_{\text{MSS}}$ when $f_{\text{T}}$ (T) and $f_{\text{MSS}}$ (S) are jointly trained together. Similar to Table 2, the prefix 'p-' represents that the transcription module is pretrained. In this study, we also assume that the ground truth instruments are the target instruments that human users intended to transcribe as in Table 2.

Three merging modes between the spectrograms $X_{\text{STFT}}$ and the transcription features $g$ are studied (Figure 4). In the 'sum' mode, the $X_{\text{STFT}}$ after the batch normalization layer simply adds to $g$. The resulting merged tensor has the same dimension as $X_{\text{STFT}}$ and $g$. In the 'concat' mode, $X_{\text{STFT}}$ and $g$ are concatenated together forming a tensor with frequency dimension double the size of $X_{\text{STFT}}$ and $g$. The 'concat' mode generally performs better than the 'sum' in exchange of a higher computation complexity. We also explored a 'spec patch' mode where $g$ is added to $X_{\text{STFT}}$ directly before the batch normalization. The model predicted mask is applied to this modified spectrogram $X_{\text{STFT}} + g$ in this merge mode. We want to know if $g$ can be used to enhance the spectral features in $X_{\text{STFT}}$ corresponding to the target instrument $i$. It turns out this merging mode performs the worst among the three modes. Since the difference in SDR is not very significant between the 'sum' and 'concat' modes, we stick to the 'sum' mode.

In Table 2, it is shown that when using the ground truth piano roll, which is binary in nature, the source separation performance improved by a large margin. To understand whether the binary nature of piano roll or the fully accurate transcribed notes contribute to the boosted performance, we also experimented with two forms of transcription output: posteriorgram $Y_{\text{frame}}^{i} \in [0, 1]$ and piano roll $Y_{\text{roll}}^{i} \in \{0, 1\}$. When converting $Y_{\text{frame}}^{i}$ into $Y_{\text{roll}}^{i}$, thresholding is required, which destroys the gradient information in it. To keep the gradient information, we apply straight-through estimation (STE) Yin et al. (2019); Le et al. (2022) as the thresholding method to preserve the gradient information. The results indicate that using binary piano rolls $Y_{\text{roll}}^{i}$ has no advantage over the posteriorgrams $Y_{\text{frame}}^{i}$.

# G   EXPERIMENTAL SETUP FOR DOWNBEAT, CHORD, AND KEY ESTIMATIONS

For audio processing, we use 6-channel harmonic spectrograms (128 frequency bins) from Won et al. (2020a). We simplify the piano rolls produced by Jointist into two channels (instrument index 0-37 as channel 0 and index 38 as channel 1). We then use a 1-D convolution layer to project the piano rolls into the same frequency dimension as the spectrograms. The resulting hybrid representation is a concatenation of both the spectrograms and piano rolls, i.e., 8-channel.

SpecTNT (Lu et al., 2021), a state-of-the-art Transformer architecture, was chosen for modeling the temporal musical events in audio recordings (Hung et al., 2022; Wang et al., 2022). The times-tamp and label annotations are converted into temporal activation curves for the learning targets for

SpecTNT as done in Lu et al. (2021); Hung et al. (2022). Table 10 below summarizes our SpecTNT configurations for the three tasks.

We train the beat and downbeat tracking tasks jointly following Hung et al. (2022), but focus on downbeat evaluation, which is more challenging than beat tracking (Böck & Davies, 2020; Durand et al., 2016). We consider 24 classes of major and minor triad chords for chord estimation, and 24 classes of major and minor keys for key estimation, plus a "none" class for both tasks. The key and chord tasks are trained and evaluated separately.

For downbeat tracking, we use 7 datasets: Ballroom (Krebs et al., 2013), Hainsworth (Hainsworth & Macleod, 2004), SMC (Gouyon, 2006), Simac (Holzapfel et al., 2012), GTZAN (Marchand & Peeters, 2015), Isophonics (Mauch et al., 2009), and RWC-POP (Goto et al., 2002). We use both the Isophonics and RWC-POP for evaluation while the remaining 6 datasets are used for training. For chord estimation, we use Billboard (Burgoyne et al., 2011)[3], Isophonics, and RWC-POP. We use both Isophonics and RWC-POP for evaluation while the remaining 2 datasets are used for training. For key estimation, we use Isophonics for evaluation and Billboard for training.

| Task | Input length | $(k, d)$ | $(h_k, h_d)$ |
|---|---|---|---|
| Downbeat | 6 seconds | (128, 96) | (4, 8) |
| Chord | 12 seconds | (64, 256) | (8, 8) |
| Key | 36 seconds | (128, 32) | (8, 4) |

Table 10: SpecTNT parameters we used in each task, where $k$ and $d$ denote spectral and temporal feature dimensions; while $h_k$ and $h_d$ represent the number of heads for the spectral and temporal Transformer encoders, respectively.

---

[3]Due to missing audio files for Billboard, we redownloaded the missing files from the Internet.

| MIDI Index | MIDI Instrument | MIDI Program | Our Mapping | Our Index |
|---|---|---|---|---|
| 0-3 | Piano | Grand/Bright/Honky-tonk Piano | Piano | 0 |
| 4-5 | Piano | Electric Piano 1-2 | Electric Piano | 1 |
| 6 | Piano | Harpsichord | Harpsichord | 2 |
| 7 | Piano | Clavinet | Clavinet | 3 |
| 8-15 | Chr. Percussion | Celesta, Glockenspiel, Music box, Vibraphone, Marimba, Xylophone, Tubular Bells, Dulcimer | Chr. Percussion | 4 |
| 16-20 | Organ | Drawbar, Percussive, Rock, Church, Reed Organ | Organ | 5 |
| 21 | Organ | Accordion | Accordion | 6 |
| 22 | Organ | Harmonica | Harmonica | 7 |
| 23 | Organ | Tango Accordion | Accordion | 6 |
| 24-25 | Guitar | Acoustic Guitar (nylon, steel) | Acoustic Guitar | 8 |
| 26-31 | Guitar | Electric Guitar (jazz, clean, muted, overdriven, distorted, harmonics) | Electric Guitar | 9 |
| 32-39 | Bass | Acoustic/Electric/Slap/Synth Bass | Bass | 10 |
| 40 | Strings | Violin | Violin | 11 |
| 41 | Strings | Viola | Viola | 12 |
| 42 | Strings | Cello | Cello | 13 |
| 43 | Strings | Contrabass | Contrabass | 14 |
| 44 | Strings | Tremolo Strings | Strings | 15 |
| 45 | Strings | Pizzicato Strings | Strings | 15 |
| 46 | Strings | Orchestral Harp | Harp | 16 |
| 47 | Strings | Timpani | Timpani | 17 |
| 48-51 | Ensemble | Acoustic/Synth String Ensemble 1-2 | Strings | 15 |
| 52-54 | Ensemble | Aahs/Oohs/Synth Voice | Voice | 18 |
| 55 | Ensemble | Orchestra Hit | Strings | 15 |
| 56 | Brass | Trumpet | Trumpet | 19 |
| 57 | Brass | Trombone | Trombone | 20 |
| 58 | Brass | Tuba | Tuba | 21 |
| 59 | Brass | Muted Trumpet | Trumpet | 19 |
| 60 | Brass | French Horn | Horn | 22 |
| 61-63 | Brass | Acoustic/Synth Brass | Brass | 23 |
| 64-67 | Reed | Soprano, Alto, Tenor, Baritone Sax | Saxophone | 24 |
| 68 | Reed | Oboe | Oboe | 25 |
| 69 | Reed | English Horn | Horn | 22 |
| 70 | Reed | Bassoon | Bassoon | 26 |
| 71 | Reed | Clarinet | Clarinet | 27 |
| 72 | Pipe | Piccolo | Piccolo | 28 |
| 73 | Pipe | Flute | Flute | 29 |
| 74 | Pipe | Recorder | Recorder | 30 |
| 75-79 | Pipe | Pan Flute, Blown bottle, Shakuhachi, Whistle, Ocarina | Pipe | 31 |
| 80-87 | Synth Lead | Lead 1-8 | Synth Lead | 32 |
| 88-95 | Synth Pad | Pad 1-8 | Synth Pad | 33 |
| 96-103 | Synth Effects | FX 1-8 | Synth Effects | 34 |
| 104-111 | Ethnic | Sitar, Banjo, Shamisen, Koto, Kalimba, Bagpipe, Fiddle, Shana | Ethnic | 35 |
| 112-119 | Percussive | Tinkle Bell, Agogo, Steel Drums, Woodblock, Taiko Drum, Melodic Tom, Synth Drum | Percussive | 36 |
| 120-127 | Sound Effects | Guitar Fret Noise, Breath Noise, Seashore, Bird Tweet, Telephone Ring, Helicopter, Applause, Gunshot | Sound Effects | 37 |
| 128 | Drums | Drums | Drums | 38 |

Table 11: The instrument mapping used in our experiments. Our mapping is less detailed than the MIDI Program Number, but it is finer than the MIDI Instrument code, thus resulting in 39 different instruments.

