# OpenReview forum: "Jointist: Simultaneous Improvement of Multi-instrument Transcription and Music Source Separation via Joint Training"
_ICLR.cc/2023/Conference — Submitted to ICLR 2023_

### Official Review · Reviewer_jSBB · 2022-10-24

**Confidence:** 3
**Correctness:** 3
**Technical Novelty And Significance:** 2
**Empirical Novelty And Significance:** 2
**Recommendation:** 6

**Clarity, Quality, Novelty And Reproducibility:**

The paper is written clearly, is original albeit not revolutionary, and is most likely reproducible modulo the standard fiddling with hyperparameters and whatnot.

**Strength And Weaknesses:**

Strengths
-----------
There is one main strength which is that the overarching goal of the paper is solid: solve transcription and source separation jointly for music with an arbitrary number of instruments.  And the system proposed in the paper does appear to be a step forward in solving both problems, as demonstrated by the many experiments in the paper.

Weaknesses
--------------
1) It's not especially surprising that combining transcription and source separation would lead to improved performance on both tasks.  However, there are multiple ways one could imagine combining these tasks.  In this paper, transcription seems to be treated as "primary", as the output of the transcription module is fed into the source separation module.  One could just as well imagine the converse, where sources are separated prior to transcription.  Would that setup work better?  Or could there be a third setup that learns a representation useful for both tasks?  These questions are neither asked nor answered by the paper.

2) In terms of the objective metrics, the transcription model in this paper performs *considerably* worse than the MT3 model of Gardner et al.  Since the purpose of music transcription is usually not direct listening, the fact that listeners prefer the Jointist transcriptions is interesting but not especially compelling.  Since you are already performing some experiments with the transcriptions on downstream tasks, it would be helpful to demonstrate that the transcriptions from Jointist also outperform MT3's transcriptions on these tasks, if that is in fact the case.

3) Related to the above, I would be interested in reading an entire paper comparing downbeat/chord/key estimation from audio vs. pianoroll.  But the fact that giving a model access to both pianoroll and audio outperforms just audio is not surprising, nor is the performance improvement large enough to warrant the inclusion of the experiment in *this* paper.  Downbeat/chord/key estimation is an application of *any* transcription model!

4) The introduction suggests a lack of symbolic data for training generative pop music models as a motivation for music transcription.  However, the very dataset used in this paper (Slakh) is derived from a much larger dataset of symbolic music (Lakh MIDI Dataset), much of which is pop.  And there are existing generative models (e.g. MuseNet) trained on such datasets.  I would argue that what's lacking is symbolic data *aligned to real recordings*.

5) Minor issues:
    * Throughout the paper the term "ablation" is used to refer to experiments with multiple conditions where none is obviously an ablation; "ablation" refers to *removal* of components.
    * Some typos: "differnt" in Section 3.3, "black" instead of "block" in Appendix A, "marco" instead of "macro" in Appendix B.1.
    * Figures 6 and 7 would be much more readable if axis labels used linear label counts even if the dimension is log scale.
    * More of a question: in Appendix E, why do some pairs of instruments that seem similar (e.g. recorder and flute) have vastly different transcription performance?  It might be helpful to show something like a confusion matrix here (even though the values aren't exactly classification counts), because I suspect there are pairs of instruments that are frequently mistaken for one another.

**Summary Of The Paper:**

The paper describes a jointly-trained triplet of models for musical instrument recognition, music transcription, and music source separation, able to handle a large number of instruments.  Experiments demonstrate that training these models jointly results in better performance for each task than training each model in isolation, and the models (the transcription model specifically) is useful for several downstream tasks.  I am not as familiar with the source separation literature, but I believe another contribution of the paper is a source separation model that can work on a much wider variety of instruments.

**Summary Of The Review:**

The paper doesn't have any especially surprising findings and could explore its main idea more thoroughly (and remove some non-essential sections), but I recommend acceptance nonetheless as it appears methodologically sound and is a valid research contribution to music transcription and source separation.

---

> ### Author Response · Authors · 2022-11-08
> **We thank Reviewer jSBB for the constructive feedback and comments. Please find our reply below.**
>
> **Q:** It's not especially surprising that combining transcription and source separation would lead to improved performance on both tasks. However, there are multiple ways one could imagine combining these tasks. In this paper, transcription seems to be treated as "primary", as the output of the transcription module is fed into the source separation module. One could just as well imagine the converse, where sources are separated prior to transcription. Would that setup work better? Or could there be a third setup that learns a representation useful for both tasks? These questions are neither asked nor answered by the paper.
>
>
> **A:** This is a valid suggestion. But in this paper, we chose transcription as the primary model and used the transcription result for source separation and other tasks in section 6. But the converse of it would be an interesting future direction to explore which is outside the scope of this paper.
>
> **Q:** In terms of the objective metrics, the transcription model in this paper performs considerably worse than the MT3 model of Gardner et al. Since the purpose of music transcription is usually not direct listening, the fact that listeners prefer the Jointist transcriptions is interesting but not especially compelling. Since you are already performing some experiments with the transcriptions on downstream tasks, it would be helpful to demonstrate that the transcriptions from Jointist also outperform MT3's transcriptions on these tasks, if that is in fact the case.
>
> **A:** Note that comparing MT3 (NLP-based) directly to Jointist (piano roll-based) is already not fair to begin with due to the difference in transcription formulation. But when we compare Jointist to Omnizart (both piano roll-based), we found that Jointist is more robust to a huge set of instruments and hence outperforms Omnizart by a large margin. With the piano roll-based approach, Jointist has a hard time outperforming MT3 in terms of transcription accuracy.
> Second, the subjective evaluation is to evaluate how robust MT3 and Jointist are when dealing with pop songs in general (where the ground truth labels are not available). This study consists of 40% professional musicians and 60% amateur musicians (Figure 5), and Table 3 shows that the subjects agree that Jointist (2,93) is more accurate than MT3 (2.76) when transcribing pop music.
>
> Also NLP-based model such as MT3 has a “ghost output” problem. It transcribes non-existent notes when dealing with pop music. This causes a lower “Overall Note Accuracy” score for MT3 in the subjective evaluation.
> The problem of “ghost output” has been discussed in social media.
> https://twitter.com/lunixbochs/status/1574848911503634432?s=20&t=ZK5VKy3gLxJqp806JwIXbA
> https://twitter.com/jesseengel/status/1575138400708923399?s=20&t=5P_SMc6bA184FSb9SLWvwA
> Jointist is free from these problems and hence achieves a higher “Overall Note Accuracy” in the subjective evaluation.
>
> **Q:** But the fact that giving a model access to both pianoroll and audio outperforms just audio is not surprising,
>
> **A:** Because piano roll helps improve the source separation performance, it is natural to explore what other tasks would piano roll also help. Although the result is not surprising, it is still an under-explored area.
>
> **Q:** The introduction suggests a lack of symbolic data for training generative pop music models as a motivation for music transcription. However, the very dataset used in this paper (Slakh) is derived from a much larger dataset of symbolic music (Lakh MIDI Dataset), much of which is pop. And there are existing generative models (e.g. MuseNet) trained on such datasets. I would argue that what's lacking is symbolic data aligned to real recordings.
>
> **A:** Thanks for pointing out our wording error. Yes, we meant the lack of symbolic data aligned with real recordings and we have updated our paper.
>
> **Q:** Some typos
>
> **A:**  We have fixed the typos
>
> **Q:** Figures 6 and 7 would be much more readable if axis labels used linear label counts even if the dimension is log scale.
>
> **A:**  We have updated the figures, and improved the readability.
>
> **Q:** Why do some pairs of instruments that seem similar (e.g. recorder and flute) have vastly different transcription performance?
>
> **A:** There are two reasons. First, although recorder and flute might sound similar to amateur listeners, their sounds are distinctive. So it makes sense that the transcription performances are different.
> Second, as shown in Figure 7, there is only one audio sample containing a recorder in the test set of the Slakh2100 dataset. While there are much more flute samples in the test set, so transcribing one sample correctly is much easier than transcribing multiple samples correctly.

---

### Official Review · Reviewer_ZUz4 · 2022-10-25

**Confidence:** 5
**Correctness:** 1
**Technical Novelty And Significance:** 1
**Empirical Novelty And Significance:** 2
**Recommendation:** 3

**Clarity, Quality, Novelty And Reproducibility:**

The paper is hard to read with a lot (from my opinion too many) bibliographical references which are not always adequate.
The original contributions except the combination of 3 existing deep neural network is not highlighted.
Results are limited but all are reproducible.

**Strength And Weaknesses:**

Strength:
-Reproducible results with public code and audio samples

Weaknesses:

-Lack of novelty (reuse 3 existing systems trained separately on the 3 addressed tasks)
-Details of the incremental contributions and methods are not sufficient
-insufficient validation (based on only a unique synthesized dataset Slakh2100)
-The interest of combining simultaneously 3 systems appears to be limited according to results

**Summary Of The Paper:**

This paper proposes an end-to-end method so-called ``Jointist'' to simultaneously address 3 audio tasks consisting respectively
in separating the source signals, transcribing the music and recognizing the instruments.

The proposed method is a deep neural network which combines together 3 existing neural network architectures which are
respectively [Won et al. 2021] for instrument recognition, [Hawthorne et al. 2017] for the transcription and
[Jansson et al. 2017] for the source separation.

**Summary Of The Review:**

The main contribution of the paper is the combination of 3 existing deep learning-based methods into a unique end-to-end system for transcription / instrument recognition and separation.

Despite it could be interesting from a practical point of view to investigate how instrument recognition can help to improve transcription  and separation, the authors failed to convince a reader of the interest of their work for the reasons listed below:

1)from a practical point of view, the experiments are insufficient and don't always compare the proposed framework with the relevant state of the art method for each distinct task (eg. source separation results only present the method of the author(s)). Moreover, there exist several other research datasets for music transcription, instrument recognition and source separation which should also be investigated (at least one other reference dataset for each addressed task with the corresponding state-of-the-art method instead of using the same slakh2100 dataset).

2) Numerical results are limited and not sufficiently detailed. For example, Recall and Precision and confusion matrices are common metrics that could be used for the Instrument Recognition. There also exist other objective metrics for source separation such as SIR and SAR (cf. BssEval) but also perceptual-relevant metrics such as PEASS which could be used to present the separation results.

3)Important details about the proposed method are missing. It is not clear how the 3 modules can be simultaneously trained. The fig. 1 is insufficient to describe the overall method and don't explain how the 3 networks  are connected together.

To conclude, I think that this paper needs an overall reorganization to be consistent and self-content while highlighting the original contribution from the author(s).
Each experiment should be conduced more rigorously and should provides arguments about the relevance of combining the 3 systems in a unique end-to-end framework with a fair comparison with the state of the art of each distinct task.
The ideas of combining separation with transcription, and instrument recognition with transcription are not novel. Hence the authors
should further explain in what they believe that this work is original and preferable than using 3 distinct dedicated methods.

---

> ### Author Response · Authors · 2022-11-08
> **We thank Reviewer ZUz4 for the comments. We would like to clear some of the potential misunderstandings below**
>
> **Q:** Lack of novelty (reuse 3 existing systems trained separately on the 3 addressed tasks)
>
> **A:** We strongly disagree on this statement. Only the instrument recognition f_IR is a reuse of [Won et al. 2021]. f_T and f_MSS are new designs based on the model architecture of [Hawthorne et al. 2017], [Jansson et al. 2017]. More specifically, we introduced a one-hot conditioning mechanism to f_T, and double conditioning to f_MSS. Section 3.2 and 3.3 described our new designs.
>
> **Q:** from a practical point of view, the experiments are insufficient and don't always compare the proposed framework with the relevant state of the art method for each distinct task (eg. source separation results only present the method of the author(s)). Moreover, there exist several other research datasets for music transcription, instrument recognition and source separation which should also be investigated (at least one other reference dataset for each addressed task with the corresponding state-of-the-art method instead of using the same slakh2100 dataset).
>
> **A:** In this paper, we emphasize more on the transcription task. For source separation, we aim at showing the positive effect of using piano rolls as an extra condition. Achieving SOTA source separation is not our main goal and hence beyond the scope of this paper. Self-comparison with and without the piano roll condition already serves our purpose.
>
> **Q:** Numerical results are limited and not sufficiently detailed. For example, Recall and Precision and confusion matrices are common metrics that could be used for the Instrument Recognition. There also exist other objective metrics for source separation such as SIR and SAR (cf. BssEval) but also perceptual-relevant metrics such as PEASS which could be used to present the separation results.
>
> **A:** Similar to the reply above, we are not trying to achieve SOTA source separation results in this paper. Reporting SDR on Table 4 already serves our purpose of showing piano rolls improve source separation results.
>
>
> **Q:** Important details about the proposed method are missing. It is not clear how the 3 modules can be simultaneously trained. The fig. 1 is insufficient to describe the overall method and don't explain how the 3 networks are connected together.
>
> **A:** Figure 1 serves as a high-level description of how the three models are connected.
> Figure 2-4 show how each model interacts with each other. In short, f_IR outputs I_cond^(i). I_cond^(i) is used by f_T to produce transcription Y_frame. Y_frame and I_cond^(i) are used by f_MSS to produce the source.
>
> **Q:** Each experiment should be conduced more rigorously and should provides arguments about the relevance of combining the 3 systems in a unique end-to-end framework with a fair comparison with the state of the art of each distinct task.
>
> **A:** We disagree on this statement. As mentioned in one of our replies above, we are not aiming to achieve SOTA performance in all tasks in the current work. We aim at investigating the effect of joint training on both music transcription and music source separation. To prove that joint training helps, we need to compare the performance of training the model individually and jointly. Table 2 serves this purpose already.
>
> Also, the transcription module is a major part of this framework as mentioned in the second last paragraph of section 1 that we aim at "demonstrating the utility of transcription module as a pre-processing module of MIR systems". So it makes sense that we focus on evaluating the transcription module more than other modules.
>
> **Q:** The ideas of combining separation with transcription, and instrument recognition with transcription are not novel. Hence the authors should further explain in what they believe that this work is original and preferable than using 3 distinct dedicated methods.
>
> **A:** We disagree on this statement. We have mentioned in section 3.3 and 6.1 that using the output of the transcription module as the extra condition for other tasks on top of joint training is the novelty of our work. This idea makes Jointist different from existing work.
>
> **Q:** Hence the authors should further explain in what they believe that this work is original
>
> **A:** This work is original because it uses piano rolls as one of the the inputs for source separations, downbeat detection, key detection, chord detection, and music tagging. This idea is not explored in existing work in which they only consider audio as the sole input.
>
> **Q:** [Why this work is] preferable than using 3 distinct dedicated methods.
>
> **A:** As we already pointed out in the replies above, we also aim at exploring the effect of joint training in this paper. In Table 2, 4, and 5, we have proven that joint training is better than distinct dedicated methods given the current model architecture. We also mentioned in section 7 that achieving SOTA performance on all three different tasks would be our future work.

---

> > ### Comment · Reviewer_ZUz4 · 2022-12-02
> > **Thanks for the replies**
> >
> > However I am still not convinced about the significance of this contribution. Please carefully check the literature and previous work based on source separation using transcription (not only deep learning papers). My score remains unchanged.

---

### Official Review · Reviewer_YdWq · 2022-11-03

**Confidence:** 5
**Correctness:** 2
**Technical Novelty And Significance:** 2
**Empirical Novelty And Significance:** 2
**Recommendation:** 3

**Clarity, Quality, Novelty And Reproducibility:**

See above. The clarity could be improved considerably by revising the writing, adding relevant details to the main text. Reproducibility is unclear, as the authors did not state whether they plan to release open-source code to accompany the work.

**Strength And Weaknesses:**


# Major comments

* The motivation for the current work in the introduction does not, as I understand it, actually motivate the current work. The authors say that "the number of instruments in a pop song may vary....it is limiting to have a model that transcribed a pre-defined fixed number of musical instruments". However, MT3 also does not transcribe a fixed number of instruments, and transcribes the same set of outputs as the current model. I think a better motivation for this work is (1) the general trend of AI/ML models toward "foundation" models trained simultaneously to perform multiple tasks with shared parameters, and (2) a focus on pop music. The latter, in particular, seems to be a motivating factor for the current work, since the evaluation *only* focuses on such music, while the baselines (Omnizart, MT3) can transcribe other common domains as well (e.g. classical) which are absent from the evaluations.

* In several places, the characterization of Jointist vs. MT3, the current state of the art in transcription, seems unfair. For example, the abstract claims that Jointist "achieves state-of-the-art performance...outperforming ...MT3". However, Table 2 shows a clear advantage of MT3; the only improvement over MT3 is in the subjective study (see comments below). The authors also claim that "jointist is more flexible than MT3 in which human users can pick the target instruments that they are interested in"; however, one could also do this with MT3 outputs by simply retaining only the desired instruments. The latter point, in particular, is important: the ground-truth (or "human-provided)") labels are used for Jointist in many experiments (i.e. many of the entries in Table 2) and lead to the best performance; in contrast, such information could be, but is not, applied to the outputs of MT3 -- which would be a more fair comparison. This form of conditioning (essentially filtering of MT3 outputs for known instruments) would also remove the "NA" values for MT3 in Table 2, which the authors state are due to extraneous predicted instruments in the output (something avoided by their models by design in the cases where ground-truth instrument labels are provided).

* Too much material is relegated to the supplement that belongs in the main text. For example, many important architectural details and experimental results (Figs 2-4, the "new instrument-wise metric" in 4.3.1, experimental descriptions for section 6.1).

* The main improvement of the model seems to be in the human listening study. This is concerning for a few reasons. (1) The listening study results are at odds with the quantitative results in Table 2, where it seems MT3 outperforms all versions of Jointist. If the claim is that MT3 does not generalize well to new datasets, this could be verified with a similar listening study on the Slakh test set. (2) The study itself is not well-described: What are the mean and SD of the subjective scores, and how did these vary by piece and dataset? Who are the participants in the human evaluation (musical experts, non-experts)? What was the motivation for providing only MIDI files and allowing participants to render it themselves, instead of providing a pre-rendered version to control for their DAW, etc.

* The non-transcription experiments (source separation, beat tracking, downbeat/chord/key) do not seem to compare to any baselines (at least not in main text), even though baselines exist for all of these tasks (e.g. the authors cite several for source separation; for beat tracking SOTA might be the "madmom" model).

* Why is only note-offset, not note-onset-offset, F1 used in the transcription experiments?

# Minor comments

* The paper refers to such categories as "real-world popular music", but the Slakh dataset is probably not a globally-representative sample of popular music. It would be better to refer to "Western popular music" or similar (as is evidenced by a single category of "ethnic" in Table 11, which encompasses several globally-popular and distinct instruments).

* Is there a reason the arrow from f_T -> f_MSS goes into the "side" of f_MSS in Figure 1 (not where the rest of the inputs are)?

* The paper should also cite, and possibly compare to, the preceding work "Unaligned Supervision For Automatic Music Transcription in The Wild" (2022) and "A Lightweight Instrument-Agnostic Model for Polyphonic Note Transcription and Multipitch Estimation" (2022).

* P3: Please describe the denoising procedure adapted from Kong et al.

* The description of MIDI mappings in 4.1 is not clear. (1) Are all MIDI numbers 0-3 mapped to a single MIDI number (i.e. electric piano, etc. mapped to piano)? If so, how is this accounted for in the baselines? (2) How are different drum hits mapped to MIDI pitches in the model?

* Do you plan to open-source trained models and code associated with this work?

* Please describe the weighted F1 metric; I could not find a description of how it was weighted in the paper.

* "the language model in MT3 dominates the acoustic model". I believe there is only one model in MT3 (an encoder-decoder). Please clarify.

* "false positive piano rolls and false negbative piano rolls have undefined F1 scores" - please clarify. How is an entire piano roll FP or FN?

* What does the location of I_cond^(i) in Figure 4 signify? That it is input to both the encoder and decoder of U-net? Please clarify.

* Why not compare to a version that uses a frozen MT3 as the transcription model f_T, since the models are not trained together anyway?

* Table 1 suggests continued and monotonic improvements with depth; why not increase the number of layers beyond 4?

# Typos etc.

There are many typos in the paper; I list a few here but would suggest a thorough revision for clarity.

P3: "onsets-and-frames" -> onsets and frames

P3: "teacher-forced training is used during training"

P4: "differnt"

P4: "tracks per piece" -> instruments per piece (?)

P9 "in par with" -> on par with

P15 "instruments conditions" -> instrument conditions

P15 "full-fledge"

**Summary Of The Paper:**

The paper presents a novel approach to MIR tasks which uses a modular setup combining instrument recognition, transcription, and source separation models into a joint pipeline. The authors perform experiments to evaluate the proposed approach, but the results are mixed (with the only improvement being in a listening study, and quantitative metrics otherwise lower than baselines when they are compared to). Joint training of the modules hurts (rather than helps) performance. Evaluations are focused on what the authors call "real-world popular music" and are conducted on the Cerberus dataset and a set of songs extracted from other pop databases (the model only shows improvements on the latter).

**Summary Of The Review:**

Overall, the paper presents a novel take on an existing idea (training joint transcription + source separation ideas). The Jointist approach is more flexible than previously-proposed models for this task, as it can transcribe arbitrary sets of 128 MIDI instruments plus drums. However, the results seem mixed (the only clear transcription improvement is a small gain on a subjective listening study) some of the choices do not seem particularly motivated, and the characterization and use of existing models (such as MT3) seems unfair (see below) with other experiments (source separation, beat tracking, etc.) apparently missing baselines entirely. The paper also needs considerable revision; in particular, some important information is in the supplement that should be in the main text.

---

> ### Author Response · Authors · 2022-11-08
> **Reply Part 2**
>
> **Q:** Why not compare to a version that uses a frozen MT3 as the transcription model f_T, since the models are not trained together anyway?
>
> **A:** We have a feeling that Reviewer YdWq has some misunderstandings about MT3 and our Jointist.
> First, as mentioned in the first paragraph of Section 2, MT3 outputs a series of tokens instead of piano rolls. Our f_MSS takes piano rolls as the input, and hence MT3 is not compatible with our Jointist. But Reviewer YdWq is right that f_T can be replaced by any piano roll-based models that transcribe instrument-wise piano rolls. But the SOTA piano roll-based model, Omnizart, performs poorly when the number of instruments scales up.
> Second, f_T and f_MSS are trained together as mentioned in the last paragraph of Section 3. Only the f_IR module is trained separately.
>
> **Q:** There are many typos in the paper; I list a few here but would suggest a thorough revision for clarity.
>
> **A:** We have gone through the whole paper and fixed the typos.
>
> Based on the comments, we have concerns that Reviewer YdWq has misunderstanding about Jointist and MT3, and we hope that our replies would help clarifying things.

---

> > ### Comment · Reviewer_YdWq · 2022-11-08
> > **Token -> piano roll conversion**
> >
> > Can you clarify what is the practical difference between the output of a model such as MT3 and the current model ("MT3 outputs a series of tokens instead of piano rolls")? Perhaps I am misunderstanding, but Figure 2 of MT3 (https://arxiv.org/abs/2111.03017) states that "[o]utput tokens using the same vocabulary can be deterministically decoded back into MIDI data", which itself can be mapped to a piano roll using one of many existing libraries.

---

> > > ### Author Response · Authors · 2022-11-10
> > > **Differences between MT3 and Jointist**
> > >
> > > The piano roll output of f_T in Jointist contains gradients which is crucial for the joint training that we proposed in this paper. The output tokens of MT3 have no gradient. Even if there was a method to maintain the gradient duration tokenization, the process of decoding tokens back to piano rolls also destroys the gradient. In other words, MT3 and Jointist fundamentally treat transcription differently.
> > >
> > > Moreover, MT3 is a **model** that handles only multi-instrument transcription. While Jointist is a **framework** consisting of three different models: f_IR for instrument recognition, f_T for transcription, and f_MSS for source separation. We have shown in Table 2 and Table 4 that a jointly trained f_T and f_MSS mutually benefit each other. During inference, Jointist is completely flexible. For example, if users are interested only in the instruments appeared in the audio, they can use only f_IR to achieve the goal without running the whole framework. If they are interested in getting the violin source from the mixed audio, they can simply run f_MSS by providing the violin condition.
> > >
> > > In short, our paper focuses on a broader scope: the joint training of transcription and source separation. Whereas MT3 is just a model which specialized in transcription.
> > >
> > > Jointist is a **framework**, MT3 is a **model**.

---

> ### Author Response · Authors · 2022-11-08
> **We thank Reviewer YdWq for the comments. But we have a feeling that Reviewer YdWq might have some misunderstandings about our work. Please find our clarification below.**
>
> **Q:** The authors also claim that "jointist is more flexible than MT3 in which human users can pick the target instruments that they are interested in"; however, one could also do this with MT3 outputs by simply retaining only the desired instruments.
>
> **A:** It is not feasible to ask normal users to retrain MT3 every time to the desired instrument. Jointist has a huge advantage in this case because retraining is not needed. Users could simply provide the desired one-hot vectors to obtain the transcription for their desired instruments.
>
> **Q:** This form of conditioning (essentially filtering of MT3 outputs for known instruments) would also remove the "NA" values for MT3 in Table 2, [...] which would be a more fair comparison
>
> **A:** First, conditioning is not the same as filtering. Filtering the MT3 outputs is a post-processing procedure. Conditioning of Jointist is a built-in model feature, no post-processing is required.
>
> Second, the problem formulation of MT3 (NLP-based) is completely different from Jointist (piano roll-based), comparing Jointist to MT3 is not fair to begin with. The only fair comparison would be piano roll-based v.s. piano roll-based model. i.e. Jointist v.s. Omnizart. As shown in Table 2, Jointist outperforms Omnizart by a large margin.
>
> **Q:** What was the motivation for providing only MIDI files and allowing participants to render it themselves, instead of providing a pre-rendered version to control for their DAW, etc.
>
> **A:** We provide MIDI files to prevent participants from judging the “Overall Listening Experience” based on the pre-rendering quality. Also, the MIDI files contain all the note onset and offset information for the participants to evaluate the transcription accuracy both visually and aurally, which is not possible with pre-rendered audio.
>
> **Q:** Who are the participants in the human evaluation (musical experts, non-experts)?
>
> **A:** As shown in Figure 5 in the Appendix, it consists of 40% musical experts (composer, music producer, professional instrument player), and 60% non-experts.
>
> **Q:** What are the mean and SD of the subjective scores,
>
> **A:** The mean subjective scores are reported in Table 3. The SD of MT3 for the four scores are [1.04, 1.04, 1.05, 1.06]. The SD of Jointist for the four scores are [0.96, 1.03, 1.02, 1.03]. We have already added this information to the paper.
>
> **Q:** Why is only note-offset, not note-onset-offset, F1 used in the transcription experiments?
>
> **A:** Due to the page limit and the ambitious tasks we try to achieve, it is difficult to include everything in the main text, but  we do report the note-onset-offset F1 in Figure 10 of the Appendix.
>
> **Q:** Is there a reason the arrow from f_T -> f_MSS goes into the "side" of f_MSS in Figure 1 (not where the rest of the inputs are)?
>
> **A:** There is no specific reason, we simply do not have enough space to put all the arrows on the same side.
>
> **Q:** Do you plan to open-source trained models and code associated with this work?
>
> **A:** Yes, we plan to release the source code after the reviewing process is finished.
>
> **Q:** Please describe the weighted F1 metric; I could not find a description of how it was weighted in the paper.
>
> **A:** We have already described it in equation (4). As mentioned in the same paragraph, w(i) is the weighting for the i-th instrument proportional to its label count in the test set.
>
> **Q:** "the language model in MT3 dominates the acoustic model". I believe there is only one model in MT3 (an encoder-decoder). Please clarify.
>
> **A:** The encoder is the acoustic part and the decoder is the language part. It is a well-known issue for the language-based model like MT3 and there is an ongoing discussion about this on Twitter.
> https://twitter.com/lunixbochs/status/1574848911503634432?s=20&t=ZK5VKy3gLxJqp806JwIXbA
> https://twitter.com/jesseengel/status/1575138400708923399?s=20&t=5P_SMc6bA184FSb9SLWvwA
> Our Jointist does not have this problem and that is the reason why it outperforms MT3 in the subjective evaluation. We notice that while MT3 performs well on Slakh2100, it produces a lot of “ghost” transcription when dealing with pop music in general.
>
> **Q:** What does the location of I_cond^(i) in Figure 4 signify? That it is input to both the encoder and decoder of U-net? Please clarify.
>
> **A:** Yes, I_cond^(i) is used in each convolution block of the encoder and decoder.

---

> > ### Comment · Reviewer_YdWq · 2022-11-08
> > **Important clarification: "retrain" vs "retain"**
> >
> > Thank you for the response.
> >
> > I wanted to clarify that in my review, I did not suggest to "retrain" MT3. I specifically wrote (and meant) that you could *retain* the desired instruments. I think that this needs to be taken more seriously than it is in the author response; perhaps I can clarify.
> >
> > The authors repeatedly emphasize that they can use a form of human input to exactly specify which instrument(s) are present in the track or desired in the outputs. I suggest that this can be done with other models as well, as a postprocessing step. There is no practical difference between using this human information as a conditioning "input" to the transcription (which is, in reality, ground truth information provided to the Jointist model, and not actual human estimation or labels) vs. using as an input to a postprocessing phase (where the outputs are filtered, based on this same information, to RETAIN only the desired instruments).
> >
> > In the end, both procedures (ground-truth conditioning, vs postprocessing to retain only ground-truth instruments) have the same inputs and outputs, and use the same ground-truth labeling. The postprocessing step would also be negligible in terms of compute cost. As the experiments are currently setup, some versions of Jointist have inputs that are never provided to the other models, particularly MT3, and the comparison doesn't seem fair. In any case, it doesn't tell us how the baselines would perform if they were also able to make use of this information, which is indeed possible.
> >
> > I'll add other questions/clarifications as needed, but wanted to give the authors a chance to share their thoughts on this during the response phase.

---

> > > ### Author Response · Authors · 2022-11-10
> > > **Our conclusion is based on a fair comparison between MT3 and Jointist.**
> > >
> > > Thanks for the clarification. Regarding fairness, we would like to explain a bit more about Table 2. First, we simply use the F1 scores of MT3 reported in their paper and we did not re-run their models. So we did not have a chance to apply the postprocessing step as suggested by the reviewer. But Table 2, when used correctly, contains a fair comparison between MT3 and Jointist. The ipTS(s) row and the MT3 row (Gardner et al.) have the same setting. No ground-truth labels are provided, the models figure out the instrument conditions themselves and transcribe accordingly. But by doing so, we only know the overall transcription accuracy “Flat metric”. As explained in section 4.3.1, The “Flat metric” does not tell us the instrument-wise transcription accuracy. Anyhow, this experiment shows that Jointist has a lower flat F1 score than MT3. We conclude that there must be some instruments that Jointist is poor at which drag down the “Flat F1”. But up to this point, Jointist is consistently better than Omnizart, a model that models music transcription in the same way as Jointist.
> > >
> > > To understand how good Jointist is on each instrument, we proposed two new metrics: “instrument-wise” and “piece-wise”. Directly evaluating ipTS(s) on these two metrics yields N.A. values (a more detailed explanation of this is provided in another reply). So we used pTS(s) to obtain a valid evaluation of the two metrics. As argued by Reviewer YdWq that conditioning and postprocessing/filtering are technically the same, we can be considered pTS(s) as a filtered version of ipTS(s). The results of pTS(s) yield Figure 9 and Figure 10. We found some instruments that Jointist is good at (bass) and bass at (Timpani and viola). Then we proceed to inspect the transcribed MIDI produced by Jointist and MT3 manually on a variety of mixed audio outside Slakh2100 dataset.
> > >
> > > Upon human inspections, we found that the MIDI files of Jointist contain a more accurate transcription than MT3. To confirm this, we asked 20 participants (40% professional, 60% amateur musicians) to rate the transcriptions of MT3 and Jointist (blind test) in four different aspects and reported the results in Table 3. Table 3 is a fair comparison because no ground truth label is used. This is the result that we used to conclude that Jointist is better than MT3. Up to this point, no unfair comparison is involved. Comparing pTS(s) of Jointist to MT3 might give out the illusion of an unfair comparison. But we do not use this comparison to make any conclusion.
> > >
> > > We discussed the reasons why MT3 shows a higher F1 score in Table 2 but performs worse than Jointist in Table 3 in section 5.2 of the paper. In short, MT3 might be more accurate to train and test on the same domain (in-domain), but it is less robust to out-domain mixed audio clips. While Jointist is more robust to out-domain mixed audio clips.
> > >
> > > TL;DR: Table 2 is a fair comparison between Jointist and Omnizart. Table 3 is a fair comparison between Jointist and MT3. We believe out-domain evaluation (Table 3) has a more significant meaning than in-domain evaluation (Table 2), and hence we used Table 3 to conclude that Jointist is better than MT3.

---

> > > > ### Author Response · Authors · 2022-11-10
> > > > **N.A values in Table 2.**
> > > >
> > > > If there is a violin in the ground truth, but the models fail to detect the violin at all and produce an empty piano roll for it. Then both precision and recall are 0, leading to 0/0 in F1 score which is ill-defined.
> > > >
> > > > On the other hand, if there is no violin in the ground truth (empty piano roll), but the models predict something for it, the precision and recall are also 0, learning to an ill-defined F1 score again.
> > > >
> > > > Although we can simply set F1=0 for the first case, it does not make sense for the second case. To report instrument-wise metric correctly, we need to exclude the second case by applying conditioning/filtering to Jointist (row 5) and Omnizart (row 1) to obtain the instrument-wise F1 metric. In Table 2, we conclude that Jointist is better than Omnizart as in section 5.2. To ensure a fair comparison between MT3 and Jointist, we conducted a subjective evaluation as reported in Table 3. Here, we compare the ipTS(s) version of Jointist to MT3. We conclude that Jointist is better than MT3 in Table 3. We make this conclusion based on Table 3 because a better model performance in an out-domain dataset is more valuable than an in-domain dataset (test set of Slakh2100).

---

### Official Review · Reviewer_ZLUi · 2022-11-04

**Confidence:** 3
**Correctness:** 3
**Technical Novelty And Significance:** 2
**Empirical Novelty And Significance:** 2
**Recommendation:** 6

**Clarity, Quality, Novelty And Reproducibility:**

* The paper is clearly written and well presented
* The technical novelty is limited as each individual component exists in prior work
* The author will open source the code if published

**Strength And Weaknesses:**

Strengths:
* The paper is generally well written and easy to follow
* Although many details are omitted in the main text, significant number of details such as model setup are given in the appendix
* The background and related work are well covered
* Although the idea of joint training of transcription & separation is not new for audio processing, authors made great attempt for very challenging multi-instrument music task

Weakness
* Authors could have referred to some recent work about joint speech recognition & speech separation for multi-speaker overlapped speech
* Although authors claim the proposed model reaches the state-of-the-art, the quality of separation is still poor in many cases (in the provided samples). I suspect further improvement can be obtained if authors could optimize the model architecture / training more carefully. For example, Table 1 shows deeper Transformer has the potential to perform better. There are many convolution / self-attention architectures, especially in computer vision and speech area where the authors could try to adopt
* The comparison of the proposed model and MT3 may not be strictly fair as the model size is significantly larger, the training data is different, etc.
* The quality of some plots in the appendix can be improved

**Summary Of The Paper:**

This paper proposed a joint training framework of three components - instrument recognition, transcription and source separation. The instrument recognition module is optional and can be replaced by human inputs. The joint training of the transcription and source separation module was shown to be beneficial. The author tried to work on very challenging data with many instruments playing at the same time such as pop music. Various experiments show that the proposed method is promising for several downstream tasks.

**Summary Of The Review:**

Overall, the paper explored the interesting path of joint training of music transcription and source separation. Some experiments are informative. However, the technical novelty less significant compared to the empirical novelty. More in-depth analysis of the model behavior or optimization of model architecture that may be related to the task or the joint training technique would be more interesting to the community.

---

> ### Author Response · Authors · 2022-11-08
> **We thank Reviewer ZLUi for the constructive feedback and comments. Please find our reply below.**
>
> **Q:** Authors could have referred to some recent work about joint speech recognition & speech separation for multi-speaker overlapped speech
>
> **A:** Thanks for this constructive suggestion. We added a new citation related to joint speech recognition & speech separation for multi-speaker overlapped speech in section 2, paragraph 2 as (Shi et al. 2022).
>
> Jing Shi, Xuankai Chang, Shinji Watanabe, and Bo Xu. Train from scratch: Single-stage joint
> training of speech separation and recognition. Computer Speech & Language
>
>
> **Q:** Although authors claim the proposed model reaches the state-of-the-art, the quality of separation is still poor in many cases (in the provided samples). I suspect further improvement can be obtained if authors could optimize the model architecture / training more carefully. For example, Table 1 shows deeper Transformer has the potential to perform better. There are many convolution / self-attention architectures, especially in computer vision and speech area where the authors could try to adopt
>
> **A:** It is true that we have not fully explored different model architectures. In this paper, we want to focus on the concept of joint training of transcription and source separation, and use the piano roll obtained from transcription to assist music source separation. But optimization of Jointist would be a nice future direction.
>
> **Q:** The comparison of the proposed model and MT3 may not be strictly fair as the model size is significantly larger, the training data is different, etc.
>
> **A:** Indeed, it is not fair to compare MT3 (NLP-based) and Jointist (piano roll-based) to begin with due to different problem formulations for the transcription. The only fair comparison in our study is between Jointist and Omnizart (both piano roll-based). We have shown in Table 2 that Jointist is better than Omnizart when the instrument set is huge.
>
> **Q:** The author will open source the code if published
>
> **A:** Yes, we plan to release the source code when the reviewing process is finished.

---

### Decision · Program_Chairs · 2023-01-20

**Decision:**

Reject

**Justification For Why Not Higher Score:**

Limited novelty.

**Justification For Why Not Lower Score:**

N/A

**Metareview: Summary, Strengths And Weaknesses:**

Summary: The paper presents a joint training framework for instrument recognition, transcription and source separation. The problem is challenging and the experiments have shown promising results.

Strengths: The work present a promising end-to-end solution to a challenging task.

Weaknesses: As agreed from the reviewers, the novelty of the paper is limited. There have been many existing work on joint training in other/similar fields though not exactly the same setup.